# Copine proteins are required for brassinosteroid signaling in maize and Arabidopsis

Teng Jing[1,5], Yuying Wu[1,5], Yanwen Yu[1], Jiankun Li[1], Xiaohuan Mu[1], Liping Xu[1], Xi Wang[2], Guang Qi[1], Jihua Tang[1,3], Daowen Wang[1], Shuhua Yang [2], Jian Hua [4] & Mingyue Gou [1,3] ✉

Copine proteins are highly conserved and ubiquitously found in eukaryotes, and their indispensable roles in different species were proposed. However, their exact function remains unclear. The phytohormone brassinosteroids (BRs) play vital roles in plant growth, development and environmental responses. A key event in effective BR signaling is the formation of functional BRI1-SERK receptor complex and subsequent transphosphorylation upon ligand binding. Here, we demonstrate that BONZAI (BON) proteins, which are plasma membrane-associated copine proteins, are critical components of BR signaling in both the monocot maize and the dicot Arabidopsis. Biochemical and molecular analyses reveal that BON proteins directly interact with SERK kinases, thereby ensuring effective BRI1-SERK interaction and transphosphorylation. This study advances the knowledge on BR signaling and provides an important target for optimizing valuable agronomic traits, it also opens a way to study steroid hormone signaling and copine proteins of eukaryotes in a broader perspective.

Brassinosteroids (BRs) are an important group of growth-promoting hormones found throughout the plant kingdom[1]. Genetic studies demonstrated that BRs play essential roles during nearly all phases of plant growth and development, as BR biosynthetic or signaling mutants display multiple developmental defects, such as short hypocotyls in the dark[2–5], dwarfism[6,7], abnormal leaf angle[8], and decreased crop yields[9]. Over the past two decades, tremendous progress has been made toward the understanding of the BR signaling pathway, making it one of the best understood signaling pathways in plants[1,10,11]. BRs are perceived outside of the cell by the plasma membrane-localized receptor BRASSINOSTEROID-INSENSITIVE1 (BRI1)[12,13]. In the absence of BRs, BRI1 remains in an inactive state via interaction with the inhibitory protein BRI1 KINASE INHIBITOR1 (BKI1) or

BOTRYTIS-INDUCED KINASE1 (BIK1)[14,15]. Upon BR perception, BRI1 phosphorylates BKI1, leading to its dissociation from BRI1[14]. The released BRI1 then interacts with its co-receptor BRI1-ASSOCIATED RECEPTOR KINASE1 (BAK1, also named SERK3), which works redundantly with the three other SOMATIC EMBRYOGENESIS RECEPTOR-LIKE KINASEs (SERKs) BAK1-LIKE1 (BKK1, also named SERK4), SERK1 and SERK2 to cause trans-phosphorylation between BRI1 and SERKs[16–19]. The activated BRI1–SERKs receptor complex directly phosphorylates BR SIGNALING KINASEs (BSKs) and CONSTITUTIVE DIFFERENTIAL GROWTH1 (CDG1)[20,21]. BSKs and CDG1 further activate a family of phosphatases, called BRI1 SUPPRESSOR1/BSU-LIKEs (BSU1/BSLs)[21,22], which then dephosphorylate and inactivate BRASSINOSTEROID INSENSITIVE2 (BIN2)[21–23]. BIN2 typically targets

[1]State Key Laboratory of Wheat and Maize Crop Science, Collaborative Innovation Center of Henan Grain Crops, Center for Crop Genome Engineering, College of Agronomy, Henan Agricultural University, Zhengzhou, Henan, China. [2]State Key Laboratory of Plant Physiology and Biochemistry, College of Biological Sciences, Center for Crop Functional Genomics and Molecular Breeding, China Agricultural University, Beijing, China. [3]The Shennong Laboratory, Zhengzhou, Henan, China. [4]Plant Biology Section, School of Integrative Plant Science, Cornell University, Ithaca, NY, USA. [5]These authors contributed equally: Teng Jing, Yuying Wu. ✉e-mail: mingyuegou@henau.edu.cn

BRASSINAZOLE-RESISTANT1 (BZR1) and BRI1-EMS-SUPPRESSOR1 (BES1) for phosphorylation and degradation[24,25]. When BR levels are high, BZR1 and BES1 are released from inhibition by BIN2, dephosphorylated by protein phosphatase 2A (PP2A)[2], and subsequently enter the nucleus to activate BR-responsive gene expression[26,27].

As introduced above, nearly all of the major BR signaling components were originally identified in the model plant Arabidopsis (*Arabidopsis thaliana*). However, phylogenetic studies indicate that these genes are highly conserved among monocots and dicots[1], suggesting that the classical BR signaling is conserved at least in monocots and dicots. *OsBRI1*[28], *OsBAK1*[29], *OsGSK1*[30], *OsBZR1*[31], *ZmBRI1*[8], and *ZmBZR1*[32], the orthologs of the known Arabidopsis genes in rice and maize (*Zea mays*), have been demonstrated to have conserved functions as in Arabidopsis. Other genes conservative in BR signaling among monocots and dicots remain to be identified and characterized.

Copines are a group of highly conserved proteins ubiquitiously found in various eukaryotes including *Paramecium teraurelia*, plants, *Caenorhabditis elegans*, mouse, and human[33–37]. The conserved nature of copines in different organisms suggest that they play important roles in common biological pathways, which is supported by emerging studies demonstrating their roles in plant development, defense and stress responses[38–43]. All copines are characterized by two C2 domains at their N termini and a von Willebrand A domain at their C termini[33,44]. While the C2 domains are $Ca^{2+}$-dependent phospholipid-binding domains implicated in membrane association, the A domain is thought to be involved in protein–protein interactions[34,45–47], may thus play a role in protein complex formation and signaling.

Copines were also known as BONZAI (BON) proteins in plant. In Arabidopsis, BON1/CPN1 was originally reported to be negative regulator of plant defense because the *bon1-1* and *cpn1-1* (*bon1-4*) mutants in the Columbia-0 (Col-0) accession exhibited dwarf and autoimmune phenotypes, e.g., lesion mimic cell death, upregulated *PATHOGENESIS-RELATED* (*PR*) gene expression, and increased disease resistance[34,35,38,39]. Other than *BON1*, there are two additional homolog genes *BON2* and *BON3* in Arabidopsis[39]. The autoimmune phenotypes seen in the *bon1-1* mutant were enhanced in the *bon1-1 bon3-3* double mutant and the *bon1-1 bon2-2 bon3-3* triple mutant, displaying severe growth defect or seedling-lethal phenotypes[39].

Further characterization indicated that the TIR (Toll interleukin 1 receptor)-nucleotide-binding, leucine-rich repeat (NB-LRR) (TNL) type resistance (R) gene *SNC1* (*SUPPRESSOR OF npr1-1, CONSTITUTIVE1*) was present in the Col-0 accession but not in the Ws accession and was responsible for the accession-dependent dwarf and autoimmune phenotypes of *bon1-1* mutants[38,39], and several other TNL genes also contribute to the autoimmunity triggered by loss of *BON1* and *BON3* functions[48]. Moreover, TNL-type *R* gene-mediated disease resistance was shown to be dependent on *PHYTOALEXIN DEFICIENT4* (*PAD4*), which encodes a lipase-like protein[49], with the *pad4-1* mutation fully suppressing the dwarfism of *bon1-1*, and *bon1-1 bon3-3*[38,39]. However, the *bon1-1 bon2-2 bon3-3 pad4-1* quadruple mutant still exhibited obvious growth defects even though the autoimmunity was fully suppressed[39]. Therefore, BONs' function in plant immunity might be less conserved, and their conserved intrinsic function independent of SNC1/TNLs- and PAD4-mediated autoimmunity needs to be uncovered. For years, the attempt to identify such function has been impeded by gene redundancy of *BON* members and the existence of those TNL-type *R* genes in Arabidopsis.

Here, by studying the *BON* homologous genes in maize, we discovered that the loss of *ZmBON1* function led to a dwarf morphology not caused by autoimmunity but by deficient BR response. Consistently, the Arabidopsis *bon1-1 bon2-2 bon3-3 pad4-1* mutant showed compromised sensitivity to BR treatment, indicating that BON proteins function in BR signaling. Further study indicates that BON proteins are required for BRI1–SERKs protein complex formation and the subsequent transphosphorylation of BR signaling components, thus acting as critical regulatory components of BR signaling in both monocot and dicot plants.

## Results

### Loss of *ZmBON1* function leads to a dwarf phenotype in the monocot maize

Since the existence of TNL-type R proteins impedes the functional analysis of BON functioning in the dicot Arabidopsis, we sought to characterize the function of BON proteins in the monocot maize, wherein no TNL-type *R* genes exist. To this end, we screened the maize genome, and only two genes *ZmBON1* (GRMZM2G494514) and *ZmBON3* (GRMZM2G176995), which show close homology to *BON1* and *BON3* in Arabidopsis, were identified (Supplementary Fig. 1). We utilized clustered regularly interspaced short palindromic repeat (CRISPR)/CRISPR-associated nuclease 9 (Cas9)-mediated gene editing to generate *Zmbon1* and *Zmbon3* single mutants and *Zmbon1 Zmbon3* double mutants by targeting *ZmBON1* and *ZmBON3* alone or simultaneously in the maize inbred line KN5585. We obtained more than six different allelic mutant lines for each construct, with insertions or deletions of 1 to 16 bp generated near the protospacer adjacent motif (PAM) but no mutation in potential off-target sites (Supplementary Fig. 2). All *Zmbon1* mutant lines showed drastic dwarf morphology, while none of the *Zmbon3* single mutants showed obvious differences from wild-type plants (Fig. 1a, b). The *Zmbon1 Zmbon3* double mutants exhibited enhanced dwarfism compared to the *Zmbon1* mutants, indicating that *ZmBON3* functions additively with *ZmBON1*. All phenotypes were heritable after seven generations of planting in the field (Supplementary Fig. 3).

To determine whether the observed phenotype is consistent in different genetic backgrounds, we identified two EMS mutants in the B73 background for *ZmBON1* and *ZmBON3*, namely *Zmbon1-7* and *Zmbon3-7*, respectively, wherein point mutations were predicted to result in truncated proteins (Supplementary Fig. 4a, b). Like the *Zmbon1* and *Zmbon3* mutants in the KN5585 background, *Zmbon1-7* showed obvious dwarfism, while *Zmbon3-7* showed no morphological differences from the wild-type B73 (Supplementary Fig. 4c).

### The dwarfism of *Zmbon1* is not caused by autoimmunity in maize

Previous studies have indicated that the dwarfism of *bon1* in Arabidopsis was caused by *SNC1*-mediated autoimmunity, characterized by accelerated cell death (necrotic lesions) and upregulated defense gene expression[38,39]. However, we observed no necrotic lesions on the leaves of *Zmbon1-1*, *Zmbon3-1*, or *Zmbon1 Zmbon3-1* mutants (Fig. 1c). We analyzed reactive oxygen species (ROS) accumulation using 3,3'-diaminobenzidine (DAB) staining, which revealed no significant differences among KN5585, *Zmbon1-1*, *Zmbon3-1*, and *Zmbon1 Zmbon3-1* plants (Fig. 1d).

We then carried out transcriptome deep sequencing (RNA-seq) of the 5-week-old wild-type KN5585 and *Zmbon1-1* plants. Defense-related genes were not significantly enriched among genes upregulated in *Zmbon1-1* relative to KN5585 (Supplementary Fig. 5, Supplementary Data 1). We also analyzed the mutant for expression of the immune marker genes *ZmPR1* and *ZmPR5*, which had expression changes in the range of 10–100 times in the maize autoimmune mutants *Zmgb1^CR* (a mutant in a heterotrimeric G protein β subunit gene) and *lls1/les30* (*lethal leaf spot 1/lesion mimic 30*)[50–52]. We observed no significant differences in *ZmPR1* or *ZmPR5* gene expression in the *Zmbon1-1*, *Zmbon3-1*, or *Zmbon1 Zmbon3-1* mutant compared to KN5585 at different growth stages, when the *Zmbon1-1* mutant constantly showed obvious dwarf phenotype (Fig. 1e, Supplementary Fig. 6). Moreover, the level of the defense phytohormone salicylic acid (SA) did not change in the *Zmbon1-1*, *Zmbon3-1*, or *Zmbon1 Zmbon3-1* mutants (Fig. 1f). The above data indicate that the dwarf phenotype is not associated with autoimmune responses as in Arabidopsis.

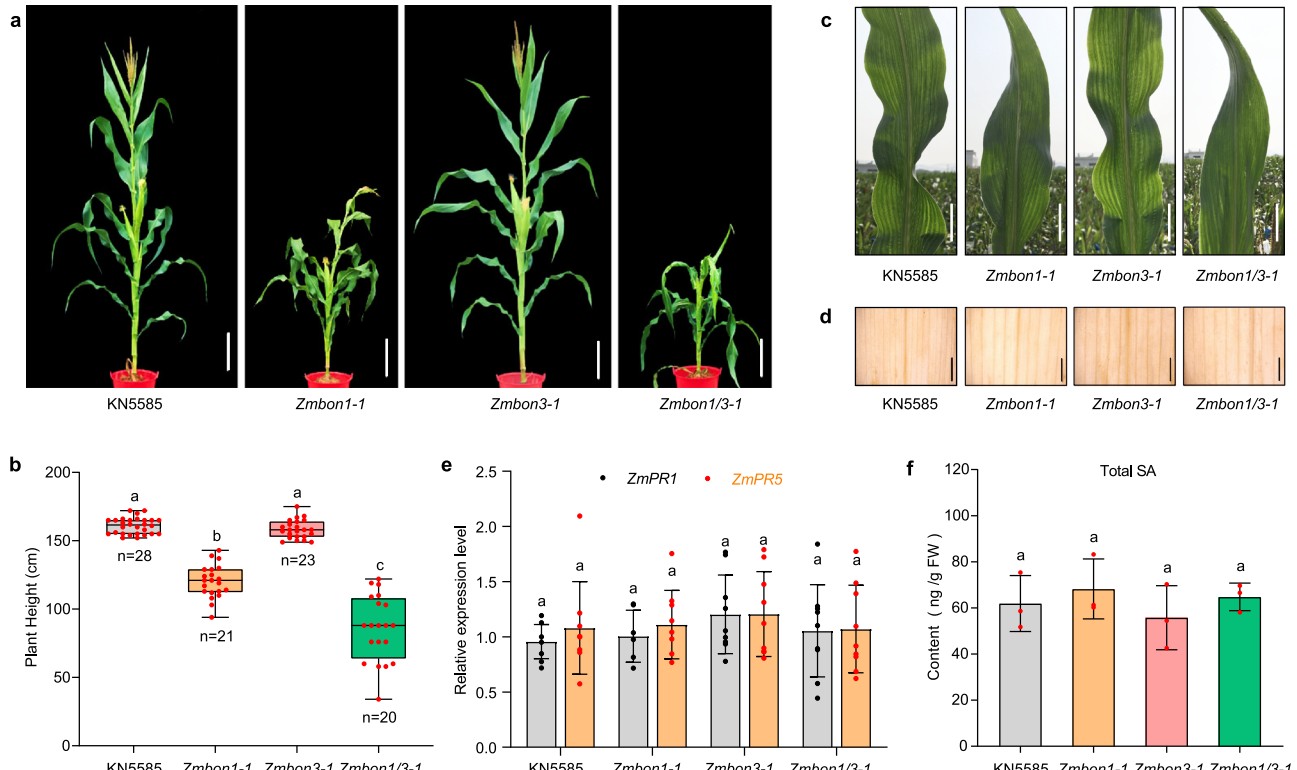

**Fig. 1 | Knocking out *ZmBON1* using CRISPR/Cas9 gene editing leads to dwarfism but no autoimmunity in maize. a** Representative images of 70-day-old wild-type KN5585, *Zmbon1-1*, *Zmbon3-1*, and *Zmbon1 Zmbon3-1* (*Zmbon1/3-1*) maize plants grown in the field in Sanya (109°17′ E, 18°35′ N). Scale bars, 20 cm. **b** Plant height of KN5585, *Zmbon1-1*, *Zmbon3-1*, and *Zmbon1 Zmbon3-1* (*Zmbon1/3-1*) plants at the mature stage. Data are displayed as box and whisker plots with individual data points. The whiskers represent maximum and minimum values, the center line represents the median and the box limits are the 25th and 75th percentiles. Red dots represent individual data points. *n* represents the number of plants. Different letters indicate significant difference between genotypes, which were determined by one-way ANOVA at *P* < 0.01. **c** Representative images of leaves from plants in (**a**). Scale bars, 5 cm. **d** 3, 3′-diaminobenzidine (DAB) staining of leaves from KN5585,

*Zmbon1-1*, *Zmbon3-1*, and *Zmbon1 Zmbon3-1* (*Zmbon1/3-1*) plants. Scale bars, 0.5 cm. **e** Relative expression levels of *ZmPR1* and *ZmPR5* in KN5585, *Zmbon1-1*, *Zmbon3-1*, and *Zmbon1 Zmbon3-1* (*Zmbon1/3-1*) plants at 4-leaf stage as determined by RT-qPCR. Black or red dots represent individual data points. Different letters indicate significant difference between genotypes, which were determined by one-way ANOVA at *P* < 0.01 (*n* = 9 biologically independent samples, ±SD). **f** Total salicylic acid (SA) content of 4-leaf stage seedlings of KN5585, *Zmbon1-1*, *Zmbon3-1*, and *Zmbon1 Zmbon3-1* (*Zmbon1/3-1*) mutants. Red dots represent individual data points. Different letters indicate significant difference between genotypes, which were determined by one-way ANOVA at *P* < 0.01 (*n* = 3 biologically independent samples, ±SD). All experiments were repeated 3 times biologically. Source data are provided as a Source data file.

## *ZmBON1* is involved in BR signaling in maize

Morphological analysis showed that the dwarfism of *Zmbon1-1* was caused by shorter rather than fewer internodes (Supplementary Fig. 7a). In addition, the leaves of *Zmbon1-1* were disordered and twisted. The leaf angle of *Zmbon1-1* had a larger variation among different leaves compared to that of the wild type (Fig. 2a), and some field-grown plants showed an obvious corkscrew appearance (Supplementary Fig. 7b). Moreover, the hundred-kernel weight was significantly reduced in *Zmbon1-1*, suggesting that the mutant produces smaller kernels (Fig. 2b). All these phenotypes are typical characteristics of maize BR-defective mutants, such as the *ZmBRI1*-RNAi lines[7,8,53–55]. We thus suspected that the growth defects of *Zmbon1* might be related to deficiency in BR signaling.

To test this hypothesis, we performed a root growth inhibition assay. Upon treatment of 100 nM 2,4-epibrassinolide (eBL), one of the most potent BRs, the root growth of KN5585 was largely inhibited. By contrast, the inhibition of root growth was significantly reduced in the *Zmbon1-1* mutant (Fig. 2c, d), indicating that BR sensitivity in *Zmbon1-1* is lower. Consistently, the *Zmbon1-1* mutant in the B73 background also showed reduced eBL sensitivity (Supplementary Fig. 8). The maize BR biosynthetic marker genes, *Brassinosteroid dependent1* (*BRD1*) and *Constitutive photomorphogenic dwarf* (*CPD*)[8], were significantly upregulated in *Zmbon1-1* as seen by reverse transcription-quantitative PCR (RT-qPCR) (Fig. 2e), another hallmark

of BR-insensitive mutants[8]. These data indicate that *ZmBON1* is involved in BR signaling in maize.

## *BON's* involvement in BR signaling is conserved in the dicot Arabidopsis

We tested whether the function of *BONs* in BR signaling is conserved between the monocot maize and the dicot Arabidopsis by conducting a root inhibition assay with the Arabidopsis mutant *bon1-1*. Exogenous application of 2 nM, 5 nM, or 10 nM eBL all significantly inhibited root growth in the wild-type Col-0, Ws, and the *pad4-1* mutant at 12 days after treatment (Supplementary Fig. 9b, c and Fig. 3b, c). The 10 nM eBL significantly inhibited root growth in both the Col-0 and *bon1-1* seedlings to a similar extent (Supplementary Fig. 9b, c). The 10-nM eBL application also equally inhibited the root growth of wild-type Ws and *bon1-2* seedlings (Supplementary Fig. 9b, c). Moreover, the *bon1-2 bon3-1* (Ws) double mutant also showed the same level of root inhibition as its wild-type Ws upon eBL treatment (Supplementary Fig. 9b, c). We further analyzed the *pad4-1* and *bon1-1 bon2-2 bon3-3 pad4-1* mutants, as the autoimmunity-conferred lethality phenotype of the *bon1-1 bon2-2 bon3-3* triple mutant is largely suppressed by the *pad4-1* mutation (Fig. 3a, Supplementary Fig. 9a). There was no difference in the degree of inhibition between *pad4-1* and Col-0 seedlings upon eBL treatment, indicating that *PAD4* is not involved in BR signaling (Supplementary Fig. 9b, c). Intriguingly, eBL treatment caused a dramatically weaker root

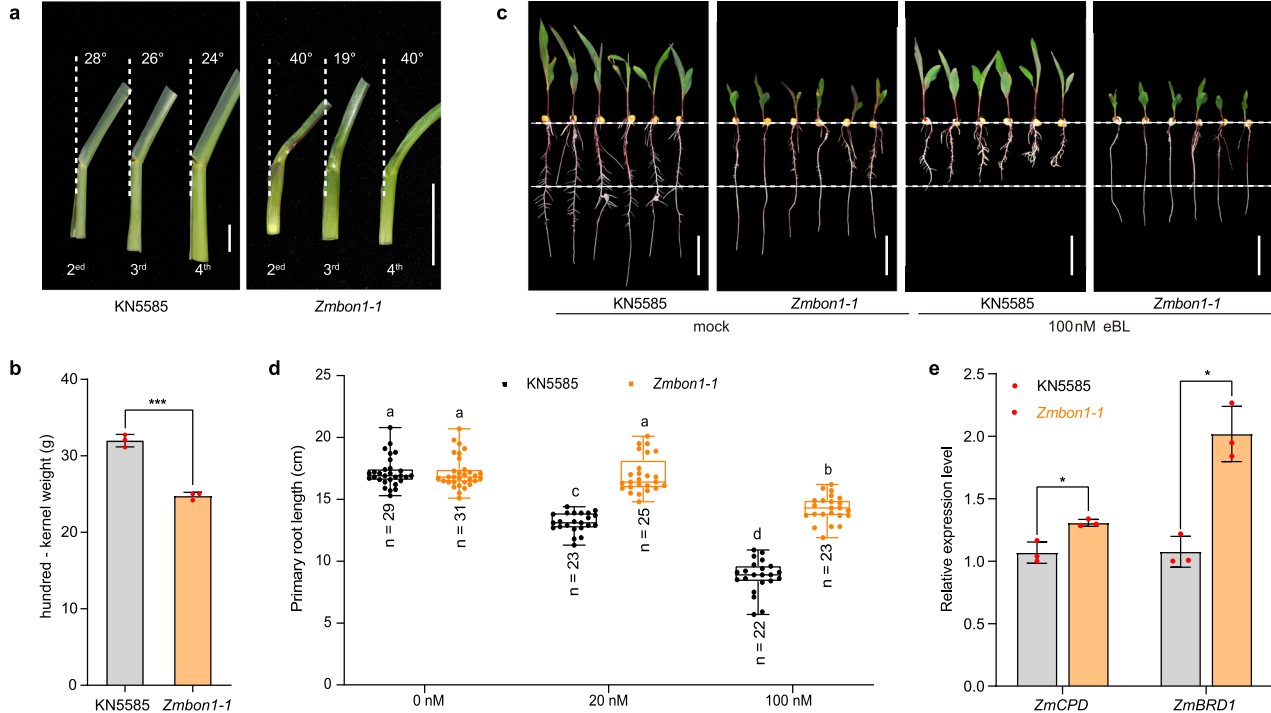

**Fig. 2 | *ZmBON1* is involved in BR signaling in maize. a** Leaf angles of KN5585 and *Zmbon1-1*. Scale bars, 3 cm. **b** Weight per hundred kernels of KN5585 and *Zmbon1-1* mutant. Significant differences were determined by two-tailed Student's *t*-tests, Asterisks indicate statistically significant differences (\*\*\**P* < 0.001, *n* = 3 biologically independent samples, ±SD). Red dots represent individual data points. **c** Root elongation phenotypes of KN5585 and *Zmbon1-1* plants under mock and 100 nM eBL treatments. Scale bars, 5 cm. **d** Primary root length of KN5585 and *Zmbon1-1* plants under 20 and 100 nM eBL treatments. Data are displayed as box and whisker plots with individual data points. The whiskers represent maximum and minimum values, the center line represents the median and the box limits are the 25th and 75th percentiles. Colored dots represent individual data points. *n* represents the number of plants. Different letters indicate significant difference between genotypes, which were determined by one-way ANOVA at *P* < 0.01. **e** Relative expression levels of *ZmCPD* and *ZmBRD1* in KN5585 and *Zmbon1-1* at 4-leaf stage. Significant differences were determined by two-tailed Student's *t*-tests, asterisks indicate statistically significant differences (\**P* < 0.05, *n* = 3 biologically independent samples, ±SD). Red dots represent individual data points. Source data are provided as a Source data file.

inhibition in the *bon1-1 bon2-2 bon3-3 pad4-1* mutant than in the *pad4-1* mutant when the phytohormone was provided as a 2-nM (Fig. 3b, c), 5-nM and 10-nM (Supplementary Fig. 9b, c and Fig. 3c) concentration, implying a defective BR signaling in the *bon1-1 bon2-2 bon3-3 pad4-1* mutant. In addition, the *bon1-1 bon2-2 bon3-3 pad4-1* mutant displayed significantly shorter hypocotyls than the *pad4-1* mutant when grown in the dark (Fig. 3d, e), further supporting the notion of the impaired BR signaling when *BON* genes were fully knocked out in Arabidopsis, since hypocotyl elongation in the dark is dependent on active BR signaling[5].

BR perception activates dephosphorylation of BZR1, a key transcriptional regulator in BR signaling[30], which then translocates to the nucleus to regulate transcription of BR-responsive genes[27]. We thus investigated the phosphorylation status of BZR1 in the *pad4-1* and *bon1-1 bon2-2 bon3-3 pad4-1* mutants in response to eBL treatment. In *pad4-1* seedlings, the abundance of dephosphorylated BZR1 increased in a time-dependent manner upon eBL treatment (Fig. 3f). By contrast, the levels of dephosphorylated BZR1 increased less in *bon1-1 bon2-2 bon3-3 pad4-1* compared to that of *pad4-1* (Fig. 3f), indicating that downstream BR signaling is impaired in *bon1-1 bon2-2 bon3-3 pad4-1* relative to *pad4-1*.

We then carried out an RNA-seq analysis to determine whether BR signaling is defective in the *bon1-1 bon2-2 bon3-3 pad4-1* mutant. As controls, we sequenced the transcriptome of the wild-type Col-0 and the null *BAK1* mutant *bak1-4*[56]. To capture the early changes after eBL treatment, the Col-0, *bak1-4*, *pad4-1*, and *bon1-1 bon2-2 bon3-3 pad4-1* plants were treated with 0.02% [v/v] Tween-20 (mock) or 2 μM eBL containing 0.02% (v/v) Tween-20 for 30 min by spray inoculation. Kyoto encyclopedia of genes and genomes (KEGG) analysis indicated that DEGs of eBL-treated Col-0 and *pad4-1* seedlings compared to

mock-treated Col-0 and *pad4-1* seedlings are enriched in the 'brassinosteroid biosynthesis' pathway (Supplementary Fig. 10a, Fig. 4a), which indicated that the transient treatment (30 min) of high concentration eBL (2 μM) successfully induced BR response in Col-0 and *pad4-1*. The Venn diagram and heatmap showed that differentially expressed genes (DEGs, fold change ≥1.5 or ≤0.5; *P* < 0.05) between Col-0 and *bak1-4* overlap largely with DEGs between *pad4-1* and *bon1-1 bon2-2 bon3-3 pad4-1* upon eBL treatment (Fig. 4b, c). Interestingly, genes in the 'brassinosteroid biosynthesis' pathway are most significantly enriched among DEGs between *pad4-1* and *bon1-1 bon2-2 bon3-3 pad4-1* seedlings upon eBL treatment (Fig. 4d). We observed a similar enrichment for the 'brassinosteroid biosynthesis' pathway for DEGs between Col-0 and *bak1-4* seedlings upon eBL treatment (Supplementary Fig. 10b), indicating that the BR response in *bon1-1 bon2-2 bon3-3 pad4-1* seedlings is affected similarly to that in *bak1-4* seedlings. Moreover, We identified only five differentially expressed genes (DEGs, fold change ≥2 or ≤0.5; *P* <0.05) between Col-0 and *pad4-1* upon eBL treatment (Supplementary Fig. 10c), and none were annotated as being related to BRs, which is consistent with the effect of eBL on the *pad4-1* mutant and Col-0 in the root inhibition assay (Supplementary Fig. 9b,c). Interestingly, the expression levels of genes involved in BR biosynthesis, including *CPD*, *ROTUNDIFOLIA3* (*ROT3*), *BR-6-OXIDASE2* (*BR6OX2*), and *DWARF4* (*DWF4*), were downregulated after eBL treatment in Col-0 and *pad4-1*, but were either not downregulated or were downregulated to a lesser extent after eBL treatment in *bak1-4* and *bon1-1 bon2-2 bon3-3 pad4-1* seedlings (Supplementary Data 2, 3). Conversely, *PHYB-4 ACTIVATION-TAGGED SUPPRESSOR1* (*BAS1*), which encodes a protein that catabolizes BRs in Arabidopsis, was strongly induced in Col-0 and *pad4-1* seedlings and was weakly induced in *bak1-*

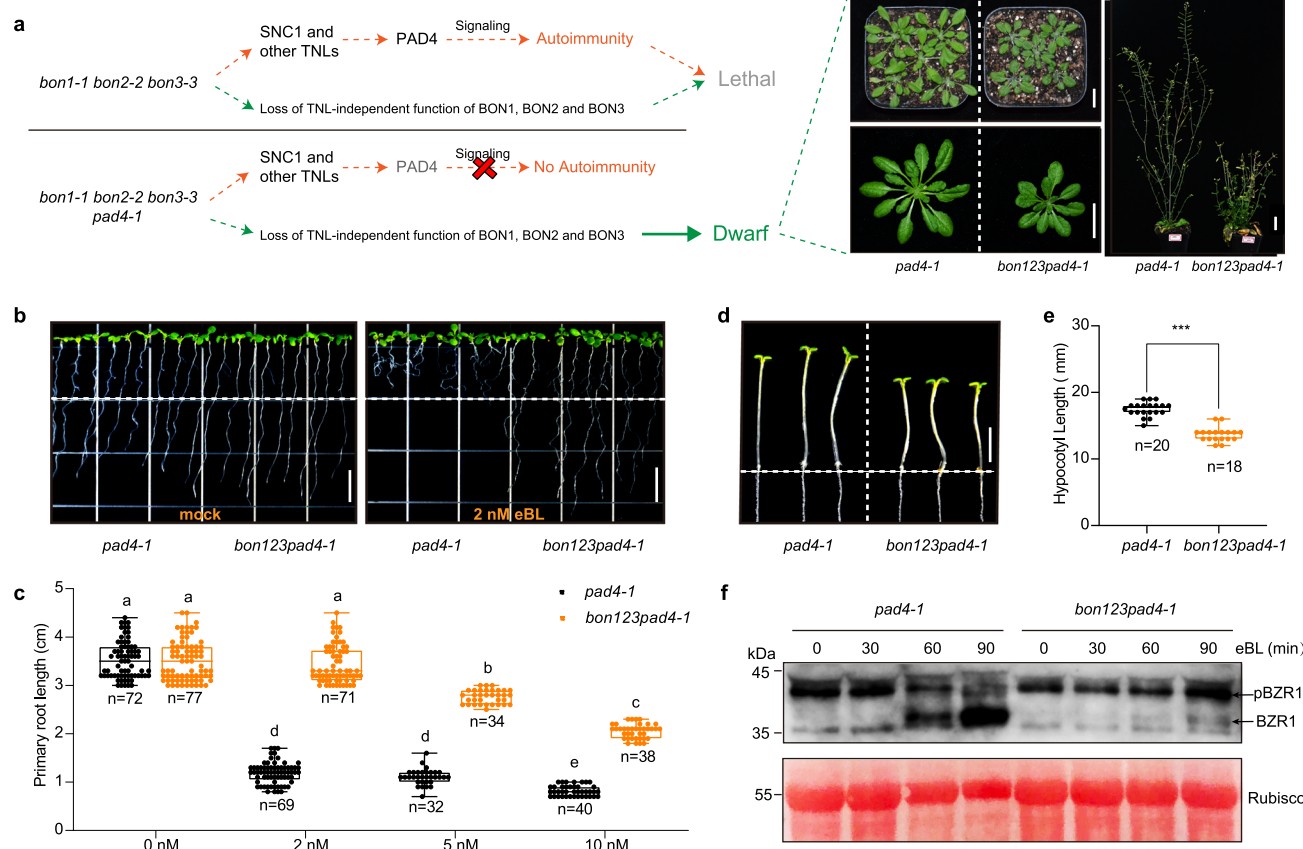

**Fig. 3 | BON proteins are involved in BR signaling in Arabidopsis. a** Schematic diagram illustrating the possible cause of the seedling-lethal phenotype of *bon1-1 bon2-2 bon3-3* and the partially rescued phenotype of *bon1-1 bon2-2 bon3-3 pad4-1*. The residual dwarf phenotypes of *bon1-1 bon2-2 bon3-3 pad4-1* (*bon123pad4-1*) plants at different growth stages are shown. Scale bar, 2 cm. **b** Root elongation phenotypes of *pad4-1* and *bon1-1 bon2-2 bon3-3 pad4-1* (*bon123pad4-1*) seedlings under mock and 2 nM eBL treatments. Scale bars, 1 cm. **c** Primary root length of *pad4-1* and *bon1-1 bon2-2 bon3-3 pad4-1*(*bon123pad4-1*) seedlings under treatment with 2, 5, and 10 nM eBL. Data are displayed as box and whisker plots with individual data points. The whiskers represent maximum and minimum values, the center line represents the median and the box limits are the 25th and 75th percentiles. Colored dots represent individual data points. *n* represents the number of plants. Different letters indicate significant difference between genotypes, which were determined by one-way ANOVA at *P* < 0.01. **d** Hypocotyl elongation phenotype of *pad4-1* and *bon1-1 bon2-2 bon3-3 pad4-1* (*bon123pad4-1*) seedlings. Scale bar, 5 mm. **e** Mean

hypocotyl length of *pad4-1* and *bon1-1 bon2-2 bon3-3 pad4-1* (*bon123pad4-1*) seedlings grown on half-strength MS medium in the dark for 5 days. Data are displayed as box and whisker plots with individual data points. The whiskers represent maximum and minimum values, the center line represents the median and the box limits are the 25th and 75th percentiles. Colored dots represent individual data points. *n* represents the number of plants. Significant differences were determined by two-tailed Student's *t*-tests, asterisks indicate statistically significant differences (***P* < 0.001). Red dots represent individual data points. **f** Immunoblot detection of BZR1 in *pad4-1* and *bon1-1 bon2-2 bon3-3 pad4-1* (*bon123pad4-1*) seedlings subjected to eBL treatment for 0, 30, 60, and 90 min. Phosphorylated (pBZR1) and dephosphorylated BZR1 (BZR1) are indicated with an arrow, respectively. Ponceau S-stained Rubisco was used as a loading control. The experiments were repeated independently 3 times with similar results. Source data are provided as a Source data file.

*4* and *bon1-1 bon2-2 bon3-3 pad4-1* seedlings after eBL treatment (Supplementary Data 3). Besides, the expression of BR-responsive gene *ARABIDOPSIS COLUMBIA SMALL AUXIN UPREGULATED GENE1* (*SAUR-AC1*) was upregulated in Col-0, *bak1-4*, *pad4-1*, and *bon1-1 bon2-2 bon3-3 pad4-1* after eBL treatment, while a mild reduction of *SAUR-AC1* expression was detected in *bon1-1 bon2-2 bon3-3 pad4-1* and *bak1-4* compared to that of *pad4-1* or Col-0 (Supplementary Data 2). In order to verify the RNA-seq data, we detected the relative transcriptional level of the six genes by RT-qPCR, which further confirmed the above data (Fig. 4e). In sum, these data collectively support the notion that *BON1*, *BON2*, and *BON3* function redundantly in regulating BR signaling in Arabidopsis.

### BONs interact with SERKs in Arabidopsis and maize
To elucidate how BON proteins regulate BR signaling, we performed a yeast two-hybrid (Y2H) screen for BON1-interacting proteins. Since the Arabidopsis BON1 is a plasma membrane-localized protein[34], and the C-terminal A domain of BON1 (BON1-A, from Val-319 to

Pro-578) is responsible for protein–protein interactions[34,46,47], we used it as a bait to screen an Arabidopsis cDNA library. We identified SERK4, which was shown to work redundantly with SERK3 in BR signaling[19], as a candidate interactor; we confirmed this interaction by retransformation in the yeast two-hybrid assay (Fig. 5a). We also independently validated the interaction between BON1 and SERK4 by a co-immunoprecipitation (Co-IP) assay in Arabidopsis protoplasts transiently transfected with the *BON1-HA* and *SERK4-FLAG* constructs (Fig. 5b) and by a split luciferase complementation (SLC) assay using a *Nicotiana benthamiana* transient expression system (Fig. 5c). Because BON1 was also previously shown to interact with SERK3[46], we tested two other members of the SERK family, SERK1 and SERK2[19], for their interaction potential with BON1. Both Y2H and SLC assays indicated that SERK1 and SERK2 also interact with BON1 (Fig. 5a, Supplementary Fig. 11). BON1 did not appear to interact with other signaling components, including BRI1, BKI1, BIK1, TETRATRICOPEPTIDE-REPEAT THIOREDOXIN-LIKE1 (TTL1), TTL4, BSK1, and BSK3, in the Y2H (Supplementary Fig. 12). Similar to the Arabidopsis proteins, ZmBON1

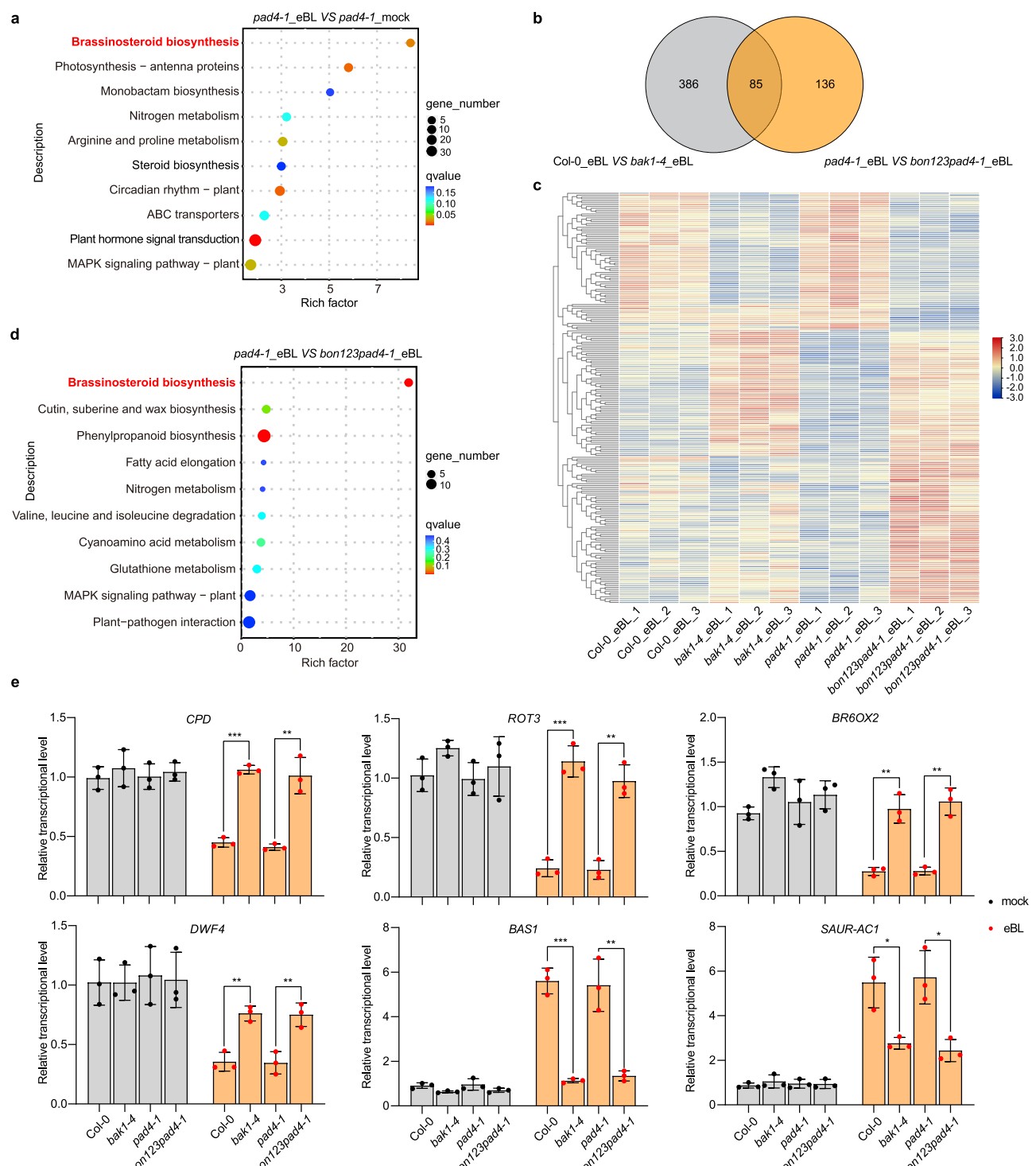

**Fig. 4 | RNA-seq analysis of Col-0, *bak1-4*, *pad4-1*, and *bon1-1 bon2-2 bon3-3 pad4-1* after mock or eBL treatment. a**, **d** Bubble chart of KEGG enrichment in DEGs in *pad4-1*_eBL vs *pad4-1*_mock (**a**) or in *pad4-1*_BL vs *bon123 pad4-1*_eBL (**d**). Each dot represents a KEGG pathway. Y-axis, pathway; X-axis: enrichment factor. A larger enrichment factor indicates a more significant enrichment of the pathway. The color of the dots indicates the *q*-value. The size of the dots represents the number of differentially expressed genes (DEGs) enriched in each pathway. **b** Venn diagram showing the number of DEGs in Col-0_eBL vs *bak1-4*_eBL and *pad4-1*_eBL vs *bon123pad4-1*_eBL. **c** Expression pattern of DEGs in Col-0, *bak1-4*, *pad4-1* and *bon1-1* *bon2-2 bon3-3 pad4-1* (*bon123pad4-1*) upon eBL treatment. The expression levels (FPKM) are normal based on the Z-core method (scale bar). **e** Relative transcriptional level of six BR-responsive genes (*CPD*, *ROT3*, *BR6OX2*, *DWF4*, *BAS1*, and *SAUR-AC1*) in Col-0, *bak1-4*, *pad4-1* and *bon1-1 bon2-2 bon3-3 pad4-1* (*bon123pad4-1*) under mock or eBL treatment as determined by RT-qPCR. Significant differences were determined using two-tailed Student's *t*-tests. Asterisks indicate statistically significant differences (*$P < 0.05$; **$P < 0.01$; ***$P < 0.001$. $n = 3$ biologically independent samples, ±SD). Black or red dots represent individual data points. Source data are provided as a Source data file.

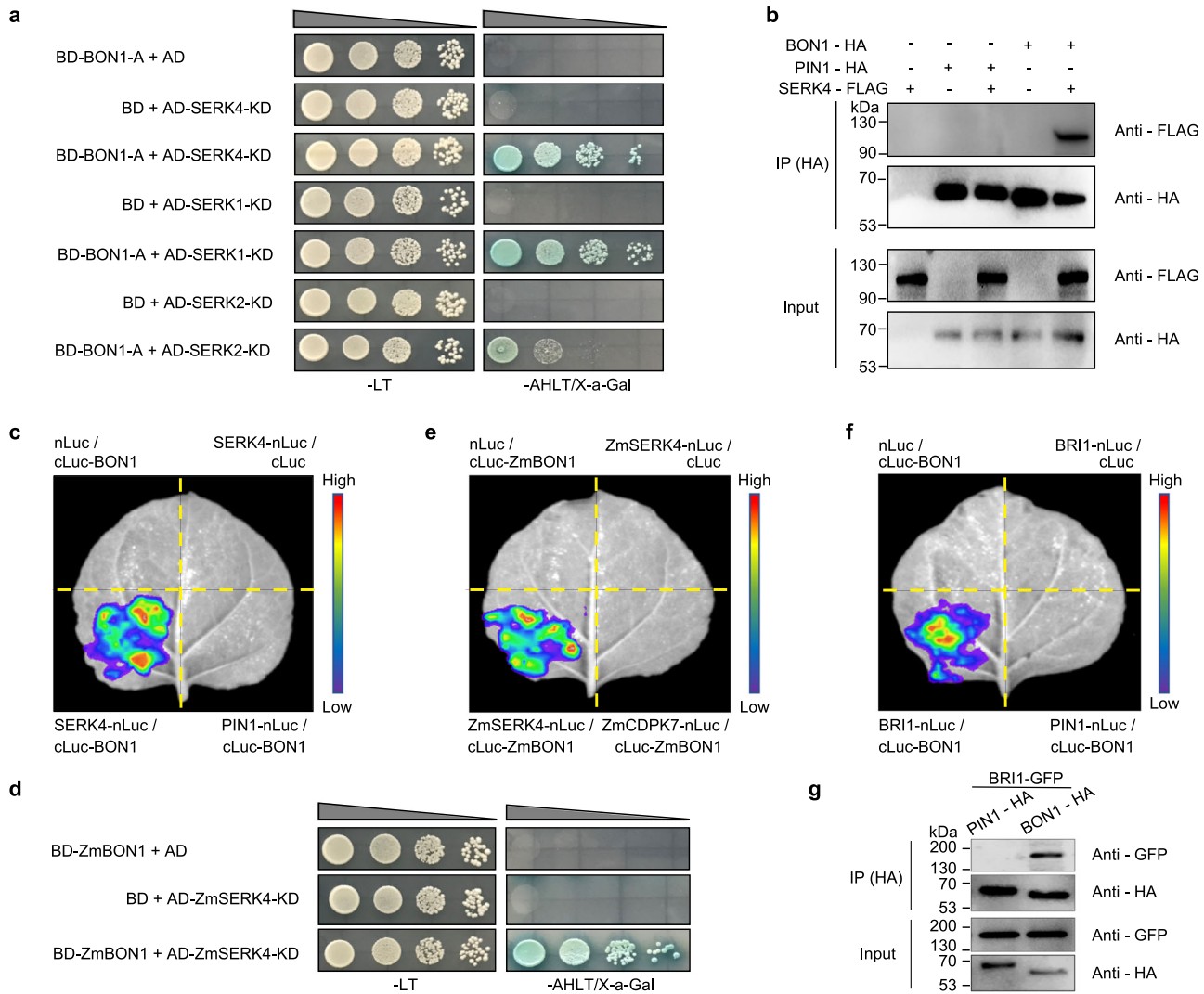

**Fig. 5 | BONs interact with SERKs and BRI1. a** Yeast two-hybrid (Y2H) assay assessing the interaction of the A domain of BON1 with the kinase domain (KD) of SERK4, SERK1 and SERK2. The constructs were co-transformed into yeast cells and grown on selective dropout medium as indicated. **b** Co-IP assay of BON1-HA and SERK4-FLAG in Arabidopsis protoplasts transfected with the respective encoding constructs. BON1-HA and SERK4-FLAG proteins were detected using anti-HA and anti-FLAG antibodies, respectively. PIN1-HA was used as a negative control. **c** Split luciferase complementation (SLC) assay showing the interaction between BON1 and SERK4. PIN1 was used as a negative control. **d** Y2H assays showing the interaction between ZmBON1 and the ZmSERK4-KD. **e** SLC assay showing the interaction between ZmBON1 and ZmSERK4 in *N. benthamiana* leaves. ZmCDPK7 was used as a negative control. **f** SLC assay showing the interaction between BON1 and BRI1 in *N. benthamiana* leaves. PIN1 was used as a negative control. **g** Co-IP assay of BON1-HA and BRI1-GFP in Arabidopsis protoplasts transfected with the respective encoding constructs. BON1-HA and BRI1-GFP proteins were detected using anti-HA and anti-GFP antibodies, respectively. PIN1-HA was used as a negative control. All experiments were repeated 3 times with similar results.

interacted with ZmSERK4 (encoded by GRMZM2G384439), a close homolog of SERK4 in maize, in the Y2H (Fig. 5d) and SLC assays (Fig. 5e), indicating that the interaction of BON1 and SERKs is conserved in Arabidopsis and maize. Interestingly, although BON1 did not appear to interact with BRI1 in Y2H assay, they associate with each other in the SLC assay (Fig. 5f) and Co-IP assay (Fig. 5g). These data suggest that BON1 and BRI1 are physically close to each other although they may not interact directly.

### BON proteins are critical for the formation of the BRI1-SERK complex

Since BON proteins directly interact with SERKs and indirectly interact with BRI1, we wonder if BON proteins may affect the stability of BRI1 and SERKs. The abundance of BRI1 and SERK3 in *pad4-1* and *bon1-1 bon2-2 bon3-3 pad4-1* were detected at 0, 30, and 60 min after eBL treatment using anti-BRI1 and anti-SERK3 antibodies (Supplementary Fig. 13a, b), the immunoblot data indicated that the abundance of BRI1

and SERK3 were not altered in *bon1-1 bon2-2 bon3-3 pad4-1* compared to that of *pad4-1* (Supplementary Fig. 14).

As the endocytosis of SERKs play important role in BR signaling[57], we then carefully examined the subcellular localization of BON1. Interestingly, other than the typical plasma membrane-localization pattern observed in previous studies[34,45], the granule signal of BON1 associated with the plasma membrane or in the cytosols was observed by switching the focal plane of the confocal (Supplementary Fig. 15a). Furthermore, the SERK4 exhibited similar localization pattern with BON1 (Supplementary Fig. 15b). Since BON1 directly interacts with SERKs, we asked whether BONs affect the trafficking of SERK proteins. However, when comparing the localization pattern of SERK3 and SERK4 in *pad4-1* and *bon1-1 bon2-2 bon3-3 pad4-1* we didn't find significant differences (Supplementary Fig. 15c, d). To further confirm BONs' effect on subcellular localization of SERK proteins, we purified the total plasma membrane proteins of *pad4-1*, *bon1-1 bon2-2 bon3-3 pad4-1*, *SERK4-FLAG/pad4-1,* and *SERK4-FLAG/ bon1-1 bon2-2 bon3-3*

*pad4-1*, and compared the abundance of SERK3 and SERK4-FLAG by immunoblots. Again, no significant difference of SERK3 or SERK4-FLAG protein level was observed (Supplementary Fig. 15e, f).

Since BON1 associate with BRI1-SERK complex, and the formation of the BRI1-SERK complex was induced by BR[16–18], we wondered if BONs might affect the interaction between BRI1 and SERKs. We generated transgenic lines expressing *SERK4-FLAG* in the *pad4-1* and *bon1-1 bon2-2 bon3-3 pad4-1* backgrounds; we chose two lines showing identical SERK4-FLAG abundance for a Co-IP assay between BRI1 and SERK4-FLAG (Supplementary Fig. 13c). The abundance of co-immunoprecipitated BRI1 increased after eBL treatment in *pad4-1*, as detected by a specific anti-BRI1 antibody (Fig. 6a, b, Supplementary Fig. 13b), consistent with previous findings that formation of the BRI1-SERK receptor complex is triggered by BR[18]. BRI1 co-immunoprecipitated with SERK4-FLAG after eBL treatment was dramatically less abundant in *bon1-1 bon2-2 bon3-3 pad4-1* than in *pad4-1* (Fig. 6a, b). In addition, using a specific anti-SERK3 antibody (Supplementary Fig. 13a), we immunoprecipitated SERK3 and compared the abundance of co-immunoprecipitated BRI1 in *pad4-1* and *bon1-1 bon2-2 bon3-3 pad4-1* seedlings. The level of co-immunoprecipitated BRI1 in *bon1-1 bon2-2 bon3-3 pad4-1* seedlings was significantly lower than that in *pad4-1* seedlings after eBL treatment (Fig. 6c, d). These data collectively indicate that BON proteins are critical for the formation of the BRI1-SERK receptor complex triggered by BR.

## Phosphorylation of BR signaling components is impaired in *bon1-1 bon2-2 bon3-3 pad4-1*

The formation of the BRI1-SERKs receptor complex contributes to the reciprocal phosphorylation of BRI1 and SERKs, which is critical for BR signal amplification[1,10,18]. To investigate the effect of impaired BRI1-SERK complex formation in the *bon1-1 bon2-2 bon3-3 pad4-1* mutant on BR signaling, the 12-day-old *pad4-1* and *bon1-1 bon2-2 bon3-3 pad4-1* seedlings treated with 0.02% [v/v] Tween-20 (mock) or 2 µM eBL containing 0.02% (v/v) Tween-20 for 30 min by spray inoculation, followed with quantitative phosphoproteomics analysis. Phosphopeptides from several known BR signaling components including BRI1, SERKs, BKI1, BSKs, and BSLs were found increased in *pad4-1* upon eBL treatment (Supplementary Fig. 16 and Supplementary Data 4), indicating that the phosphorylation cascade in BR signaling was activated in *pad4-1*_eBL. Strikingly, we detected 3.34 times more differentially abundant phosphoproteins (fold-change ≥ 2) in *pad4-1*_eBL vs *bon1-1 bon2-2 bon3-3 pad4-1*_eBL (Group A) than in *pad4-1*_mock vs *bon1-1 bon2-2 bon3-3 pad4-1*_mock (Group B) (Supplementary Fig. 16a, Supplementary Data 4), suggesting a larger difference in protein phosphorylation between the *pad4-1* and *bon1-1 bon2-2 bon3-3 pad4-1* mutants under eBL treatment. In addition, we identified 3.23 times more differentially abundant phosphoproteins in *pad4-1*_eBL vs *pad4-1*_mock (Group C) compared to the *bon1-1 bon2-2 bon3-3 pad4-1*_eBL vs *bon1-1 bon2-2 bon3-3 pad4-1*_mock comparison (Group D)

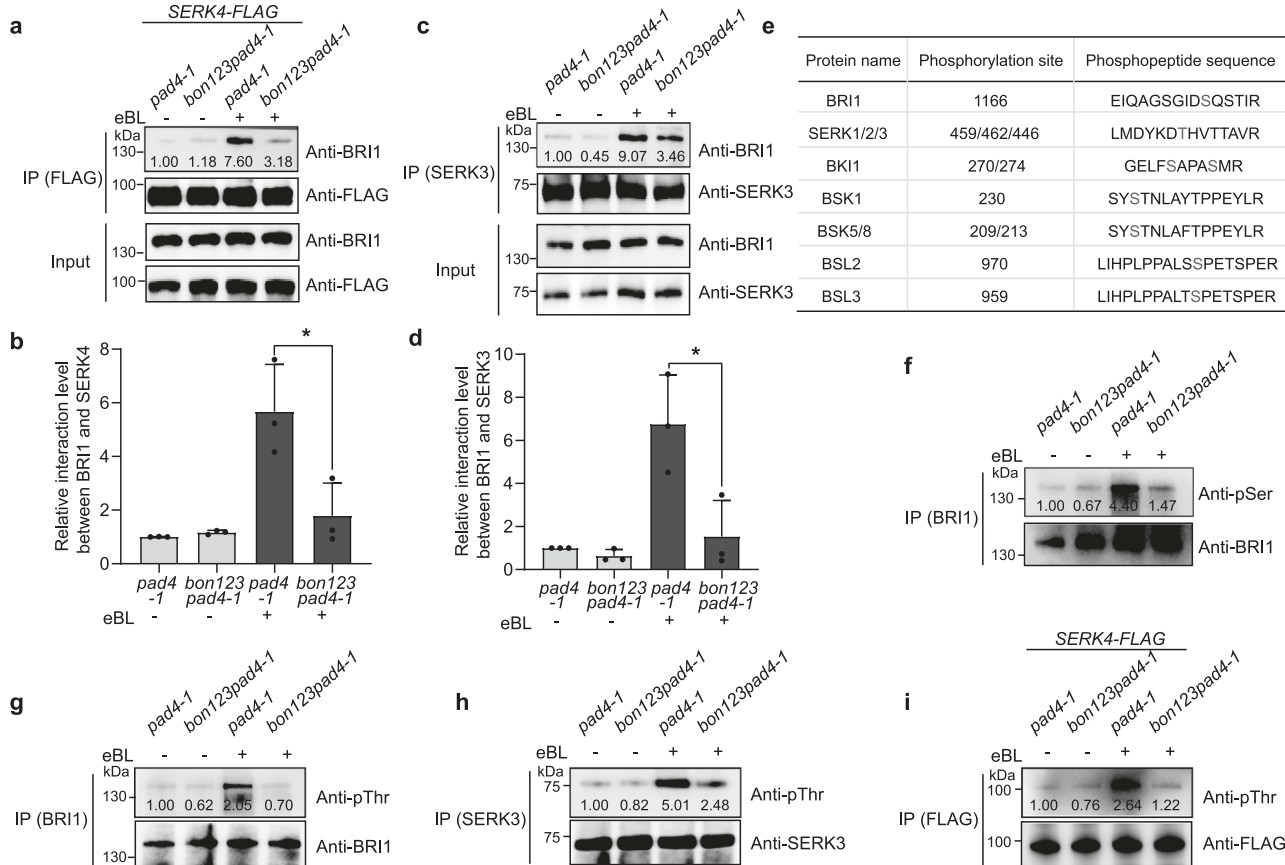

**Fig. 6 | Formation of the BRI1-SERK complex and reciprocal phosphorylation between BRI1 and SERKs are impaired in the *bon1-1 bon2-2 bon3-3 pad4-1* mutant. a, c** Co-IP assay of BRI1 and SERK4 (**a**) and of BRI1 and SERK3 (**c**) in the *pad4-1* and *bon1-1 bon2-2 bon3-3 pad4-1* (*bon123pad4-1*) mutants without or with eBL treatment. IP was performed with FLAG beads (**a**) or α-SERK3 (**c**). Immunoprecipitated SERK4-FLAG (**a**), SERK3 (**c**) and co-immunoprecipitated BRI1 were detected with specific α-FLAG, α-SERK3, and α-BRI1 antibodies, respectively. **b, d** Quantification of BRI1 associated with SERK4-FLAG (**b**) or SERK3 (**d**) shown in (**a**) and (**c**), respectively. Significant differences were determined by two-tailed

Student's *t*-tests, asterisks indicate statistically significant differences (\**P* < 0.05, *n* = 3 biologically independent samples, ±SD). Black dots represent individual data. **e** Table showing the phosphopeptide sequences and the phosphorylation sites of the BR signaling components BRI1, SERK1/2/3, BKI1, BSK1, BSK5/8, BSL2, and BSL3. Phosphorylated amino acids are shown in red. **f–i** Phosphorylation levels of BRI1 (**f, g**), SERK3 (**h**) and SERK4 (**i**) in the *pad4-1* and *bon1-1 bon2-2 bon3-3 pad4-1* (*bon123pad4-1*) mutants without or with eBL treatment. Phosphorylated proteins were detected with anti-pSer and anti-pThr antibodies. All experiments were repeated 3 times with similar results.

(Supplementary Fig. 16b, Supplementary Data 5), suggesting that the phosphorylation response to eBL was reduced in the *bon1-1 bon2-2 bon3-3 pad4-1* mutant compared to that in the *pad4-1* mutant. We only detected the phosphopeptide EIQAGSGID**S**QSTIR from BRI1 following eBL treatment, but not in mock-treated plants, and its abundance in *pad4-1*_eBL was 3.35 times that in *bon1-1 bon2-2 bon3-3 pad4-1*_eBL (Supplementary Fig. 16c, Supplementary Data 4). Interestingly, the detected phosphorylation site (S1166) was within the C-terminal region of BRI (Fig. 6e and Supplementary Data 4), which is also the specific target site of SERK3 transphosphorylation[18]. In addition, we detected the phosphopeptide LMDYKD**T**HVTTAVR, which is conserved among SERK1, SERK2, and SERK3, in the *pad4-1* mutant but not in the *bon1-1 bon2-2 bon3-3 pad4-1* mutant regardless of eBL treatment. Its abundance in *pad4-1*_eBL was 4.75 times more than that in *pad4-1*_mock, with the phosphorylation site (T446) being in the activation loop of the kinase domain, as observed in a previous study[18]. Moreover, phosphoprotein abundance of the core BR signaling components BKI1, BSK1, BSK5/8, BSL2, and BSL3, increased dramatically after eBL treatment in the *pad4-1* mutant, but was largely reduced in the *bon1-1 bon2-2 bon3-3 pad4-1* mutant after eBL treatment compared to that in the *pad4-1* mutant (Fig. 6e and Supplementary Fig. 16c), indicating that phosphorylation events of most BR signaling components are compromised in the *bon1-1 bon2-2 bon3-3 pad4-1* mutant.

To confirm the data from the above quantitative phosphoproteomics analysis, we isolated total proteins from mock- and eBL-treated *pad4-1* and *bon1-1 bon2-2 bon3-3 pad4-1* plants, and immunoprecipitated the BRI1 and the SERK proteins using their related antibodies, followed by immunoblots using the anti-phosphoserine (anti-pSer) and anti-phosphothreonine (anti-pThr) antibodies to detect the general phosphorylation changes of BRI1 and SERKs. As confirmed by three biological replicates, the general level of phosphorylated BRI1 proteins increased in *pad4-1* seedlings in response to eBL, while eBL-induced BRI1 phosphorylation was largely compromised in *bon1-1 bon2-2 bon3-3 pad4-1* seedlings relative to *pad4-1* (Fig. 6f, g and Supplementary Fig. 17a, b). We also confirmed the phosphorylation change of BRI1 in *pad4-1* vs *bon1-1 bon2-2 bon3-3 pad4-1* seedlings by detection of proteins extracted from protoplasts transiently expressing *BRI1-GFP* with four biological replicates (Supplementary Fig. 18). In addition, we co-immunoprecipitated SERK3 and SERK4-FLAG with anti-SERK3 and anti-FLAG antibodies, respectively, and detected the phosphorylated proteins using anti-pThr antibodies. The phosphorylation levels of SERK3 and SERK4-FLAG were also dramatically decreased in *bon1-1 bon2-2 bon3-3 pad4-1* seedlings compared to *pad4-1* seedlings upon eBL treatment, as confirmed by three biological replicates (Fig. 6h, i and Supplementary Fig. 17c, d), which was consistent with the results of the quantitative phosphoproteomics. The above data support the idea that BON proteins are critical regulators of the reciprocal phosphorylation between BRI1 and SERKs.

## Discussion

In this study, by characterization of the maize *Zmbon1* mutant wherein no autoimmunity is triggered since no TNL-type R genes exist in maize, we found *ZmBON1* was involved in BR signaling (Figs. 1 and 2). Moreover, the conserved function of *BONs* in BR signaling was validated using Arabidopsis *bon1-1 bon2-2 bon3-3 pad4-1* quadruple mutant wherein the *pad4-1* mutation blocked the autoimmunity and rescued the lethal phenotype of *bon1-1 bon2-2 bon3-3* (Figs. 3 and 4). While the autoimmunity of *bon* mutants in Arabidopsis is caused by the activation of TNL-type R proteins, the growth defect in maize and Arabidopsis is largely due to the deficient BR signaling. Our studies in both monocot and dicot plant systems thus uncovered the intrinsic function of *BONs* in BR signaling beyond their less conserved role in plant immunity. Since BR is also involved in growth-defense coordination[58], it will be worthy of investigation if

and how BON proteins could mediate the interplay between BR signaling and plant immunity.

Our data indicate that BON proteins are critical regulatory component for proper functioning of BR receptor complex (Fig. 7). Transcription of BR-related genes including *CPD*, *ROT3*, *BR6OX2*, *DWF4*, *BAS1*, and *SAUR-AC1*, etc. were significantly altered in *bon1-1 bon2-2 bon3-3 pad4-1* and *bak1-4* compared to *pad4-1* and Col-0 wild type upon eBL treatment (Fig. 4, Supplementary Fig. 10), these genes were also identified in previous transcriptome analysis of BR responses[59–61], implying the important role of BON proteins in BR signaling. In addition, BON proteins interacted directly with the BR co-receptor SERKs and indirectly with BRI1, and thus likely form a BON-SERK-BRI1 complex (Fig. 5). BON proteins do not appear to affect the protein stability and trafficking of BR receptor complex (Supplementary Fig. 14 and Supplementary Fig. 15). Instead, BON proteins likely act as a chaperone or structural proteins promoting the interaction between BRI1 and SERKs upon ligand binding (Fig. 6). When BON proteins were depleted, the BR-triggered interaction between BRI1 and SERKs was largely attenuated (Fig. 6). In support of the above notion, BR-triggered transphosphorylation of BRI1 and SERK proteins were largely reduced in the *bon1-1 bon2-2 bon3-3 pad4-1* mutant, as detected by both phosphoproteomics analysis and immunoblotting (Fig. 6, Supplementary Figs. 14, 16, 17, 18). The phosphorylation of BKI1, BSKs, and BSLs and dephosphorylation of BZR1 were also attenuated in the *bon1-1 bon2-2 bon3-3 pad4-1* mutant (Supplementary Fig. 16, Fig. 3). Therefore, BON proteins are involved in the regulation of entire BR signaling pathway via modulating the functioning of BR receptor complex. According to previous studies, BON1 was known to be associated with the detergent-resistant microdomains and its localization in plasma membrane is essential for BON1's function[45]. Interestingly, recent studies have found that the existence of preassembled BRI1-SERKs complexes in the microdomains of plasma membrane[62,63]. Further research is needed to investigate if BON proteins act as a structural protein affecting the microdomain formation or attracting the SERKs proteins to transport to the microdomain of plasma membrane, thereby affecting the formation of BRI1-SERKs complex. Besides, we could not exclude the possibility that dephosphorylation and/or autophosphorylation of BRI1 or SERKs are regulated by BON proteins based on current data.

Since copine proteins were ubiquitously found in different eukaryotes and BR is a steroid hormone that is shared between plants and animals, it is possible copine proteins could act as an evolutionary hub of steroid hormone signaling in different eukaryotes. This hypothesis is at least partly supported by the current finding that BONs' function in BR signaling is conserved in both monocot and dicot. This study thus opens a way for scientists from a broad life science community to explore the potentially similar function of copines in other organisms. In addition, previous structural model of BR receptor complex is based solely on the BRI1, BAK1, and BKI1[64–67]. Further structural studies on BON-SERK-BRI1 complex wait to be performed to gain deeper insights into the sophisticated functioning of BR receptor. Moreover, the pleiotropic regulatory effects of BONs render it an important target for optimizing valuable agronomic traits in crops.

## Methods

### Plant materials and growth conditions

For phenotypic observations, maize (*Zea mays*, inbred lines KN5585 and B73) plants were planted in the field in Zhengzhou (113°66′ E, 34°79′ N) in summer or Sanya (109°17′ E, 18°35′ N) in winter from 2019 to 2022. For laboratory experiments, maize plants were grown at 22 °C under long-day conditions (14-h day/10-h night) in a growth chamber. Arabidopsis (*Arabidopsis thaliana*) plants from the Col-0 and Ws accessions used in this study were grown at 22 °C under long-day conditions (14-h day/10-h night) in chambers either on potting soil or

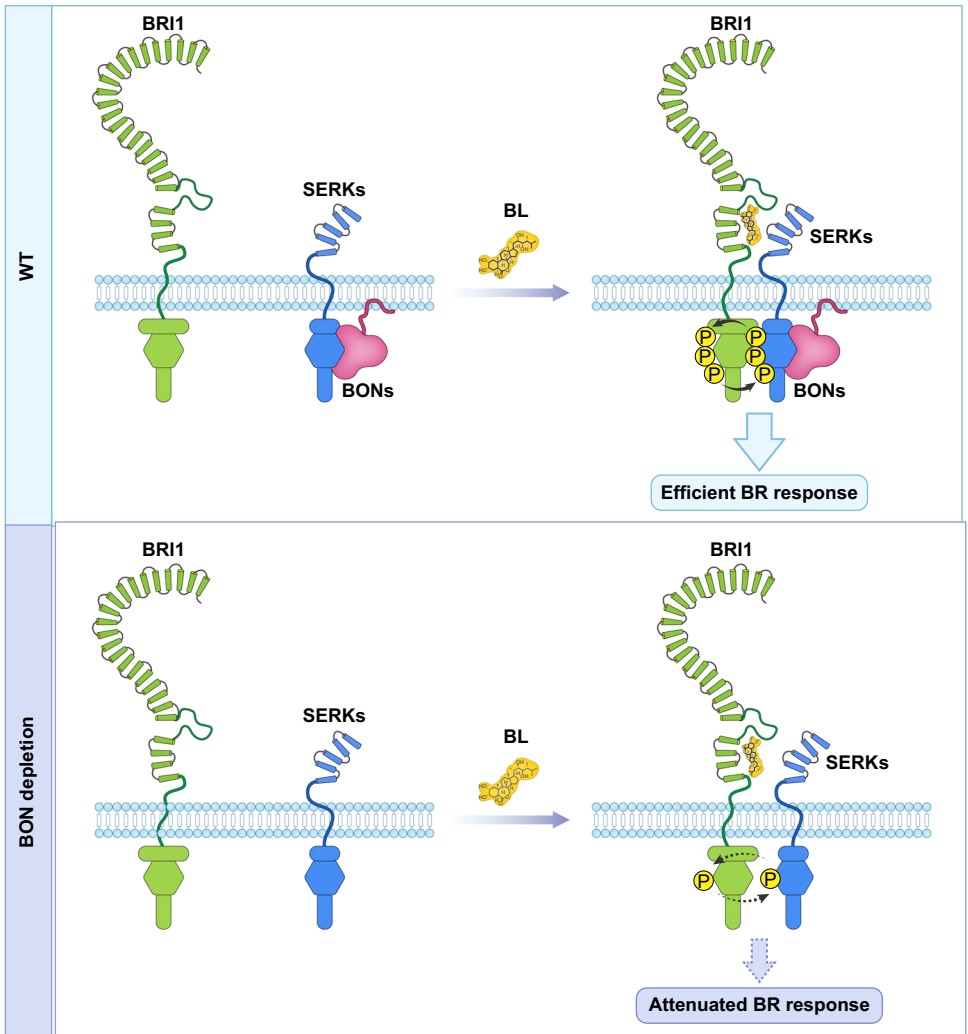

**Fig. 7 | Proposed model for the role of copine proteins in BR signaling.** In the wild type, BONs interact with SERKs, which are spatially separated from BRI1. Upon BR perception, SERKs move close to BRI1 with the assistance of BONs to form the BRI1-SERKs complex. Then, BRI1 is fully activated by reciprocal phosphorylation of BRI1 and SERKs, and the BR signal is amplified to induce an efficient BR response. In the *bon* mutants, the interactions between BRI1 and SERKs were affected, resulting in attenuated reciprocal phosphorylation between BRI1 and SERKs, leading to an attenuated BR response.

in sterile Petri dishes containing half-strength solid Murashige and Skoog (MS) medium with 2% (w/v) sucrose and 0.7% (w/v) agar. *N. benthamiana* plants were grown at 22 °C under long-day conditions (14-h day/10-h night) in chambers, and 3–4-week-old plants were used for transient expression experiments.

### Generation of mutants and transgenic lines in maize and Arabidopsis

To generate knock-out mutants of *ZmBON1* and *ZmBON3*, two single guide RNA sequences (sgRNAs) each targeting *ZmBON1* or *ZmBON3* were cloned into the pBUE411 vector[68] to generate the CRISPR/Cas9 constructs targeting *ZmBON1* and *ZmBON3* separately or simultaneously. The resulting constructs were introduced into the maize inbred line KN5585 via Agrobacterium (*Agrobacterium tumefaciens*)-mediated transformation. More than 20 independent lines were generated for each transformation. PCR products covering the target site in the respective genes were sequenced to determine the genotype of all plants. To generate *SERK4-FLAG/pad4-1* and *SERK4-FLAG/ bon1-1 bon2-2 bon3-3 pad4-1* transgenic plants, the full-length *SERK4* coding sequence was cloned into the pSuper1300-35S-3xFLAG-mCherry vector. The resulting construct was transformed into *pad4-1* and *bon1-1 bon2-2 bon3-3 pad4-1* mutant plants using the floral dip method[69].

### Reverse transcription-quantitative PCR (RT-qPCR)

Total RNA was extracted using RNAiso Plus (Takara, Cat#9109) from above-ground tissues from at least 3 maize seedlings at 4-leaf or 2-leaf stages, or from tissues of at least 50 10-day-old Arabidopsis seedlings of Col-0, *bak1-4*, *pad4-1* and *bon1-1 bon2-2 bon3-3 pad4-1* upon treatment of 0.02% [v/v] Tween-20 (mock) or 2 μM eBL containing 0.02% (v/v) Tween-20 by spray inoculation for 30 min. One microgram of total RNA was reverse transcribed into cDNA using cDNA Synthesis SuperMix (Novoprotein, Shanghai, China, Cat#E047), which was used as a template for RT-qPCR. All RT-qPCR primers used are listed in Supplementary Table 1.

### Quantification of total salicylic acid (SA)

Fresh above-ground maize tissues from 5–8 seedlings (4-leaf stage) were collected for each sample. Total SA including free SA and SA beta-glucoside (SAG) were extracted and detected by MetWare (http://www.metware.cn/) based on the AB Sciex QTRAP 6500 LC-MS/MS platform following a previously published method[70].

### BR root inhibition assay in maize and Arabidopsis

Maize seeds were soaked on sterilized wet paper towels for 2–3 days until germination. Uniformly germinated seeds were selected and transferred to layered filter paper placed vertically in a pot containing

5.0 L of deionized water alone or containing 20 nM or 100 nM of 2,4-epibrassinolide (2,4-eBL; Solarbio, Cat#SB8360). Seedlings were grown at 22 °C under a 14-h day/10-h night photoperiod for 12 days, after which primary root length was measured. Arabidopsis seeds were sterilized with 70% (v/v) ethanol and germinated vertically on half-strength solid MS medium alone or containing 2, 5, or 10 nM 2,4-eBL at 22 °C for 10 days under long-day conditions before measuring primary root length. The concentration of eBL was chosen according to previous report[4,8].

### RNA-seq analysis

For Arabidopsis, three biological replicates of 3-week-old Arabidopsis seedlings from the Col-0, *bak1-4*, *pad4-1*, and *bon1-1 bon2-2 bon3-3 pad4-1* genotypes, were mock-treated (0.02% [v/v] Tween-20) or treated with eBL containing 0.02% (v/v) Tween-20 for 30 min. For better absorption of eBL in a short time after spray inoculation, we used a higher eBL concentration (2 μM) in this study. For maize, three biological replicates of leaves from five 5-week-old plants grown in the field of Sanya were collected. All harvested plant tissues were frozen and ground in liquid nitrogen. Total RNA was isolated with RNAiso Plus (Takara, Cat#9109) and used for sequencing on an Illumina NovaSeq 6000 platform. High-quality clean data were mapped to the Arabidopsis reference genome (TAIR10). Gene expression levels were calculated as fragments per kilobase of transcript per million fragments mapped (FPKMs). Differentially expressed genes between two samples were identified using the R package DESeq (1.10.1) based on an adjusted false discovery rate (FDR) < 0.01, and a minimal absolute fold-change ≥1.5. Venn diagram and heatmap were generated using TBtools-II[71]. Statistical enrichment of DEGs in Kyoto Encyclopedia of Genes and Genomes (KEGG) pathways was performed using on the BMKCloud platform (https://www.biocloud.net)[72].

### Phosphoproteomics analysis

Four-week-old *pad4-1* and *bon1-1 bon2-2 bon3-3 pad4-1* Arabidopsis mutant plants were mock-treated (0.02% [v/v] Tween-20) or treated with 2 μM eBL containing 0.02% (v/v) Tween for 30 min. The above-ground tissues were collected, and the liquids on the surface were removed using tissue paper, then tissues of 8 individual plants were grouped as one sample and ground in liquid nitrogen. Total proteins were extracted using SDT buffer (4% [w/v] SDS, 100 mM Tris-HCl pH 7.6, 0.1 M DTT) and quantified using a BCA protein assay kit (Bio-Rad, USA). Proteins were digested by trypsin according to filter-aided sample preparation (FASP) procedure described previously[73]. Phosphopeptides were enriched using a High-Select™ Fe-NTA Phosphopeptides Enrichment Kit (Thermo Scientific, A32992), concentrated in a vacuum and dissolved in 40 μl of 0.1% (v/v) formic acid solution for mass spectrometry analysis using a timsTOF Pro mass spectrometer (Bruker) coupled to a Nanoelute (Bruker Daltonics). The data were analyzed and quantified using MaxQuant software, and the related parameters and instructions were showed in Supplementary Table 2 (APPLIED PROTEIN TECHNOLOGY).

### Protein extraction and immunoblotting

The indicated plant tissues were harvested, weighed, and ground in liquid nitrogen. Total proteins were extracted in protein extraction buffer (50 mM Tris-HCl pH 7.5, 150 mM NaCl, 10% Glycerol, 5.0 mM DTT, 2.0 mM Na₂MoO₄, 2.5 mM NaF, 1.5 mM Na₃VO₄, 0.5% [v/v] IGEPAL CA-630 [Sigma-Aldrich], 1.0 mM PMSF, 1% [v/v] protease inhibitor cocktail and 1× PhosSTOP phosphatase inhibitor cocktail [Roche]). Plasma membrane proteins were isolated using a Plasma Membrane Protein Isolation Kit (Invent, SM-005-P) according to the manufacturer's protocol. Protein samples were separated on 4–12% precast SurePAGE gels (GenScript) at 120 V for 1.5 h and transferred onto activated PVDF membranes at 200 mA for 2 h. Immunoblotting was performed using the following antibodies: anti-BZR1 (Youke

Biotechnology, Shanghai, China, YKRP082), 1:1000; anti-BRI1 (Agrisera, AS12 1859), 1:5000; anti-SERK3/BAK1 (Agrisera, AS12 1858), 1:5000; anti-FLAG (Sigma, F1804), 1:2500; anti-phosphoserine (Sigma, P5747), 1:500; anti-phosphothreonine (Cell signaling, 9381), 1:500; anti-HA (Covance, MMS-101R), 1:2500; anti-rabbit (MBL, 458), 1:10,000; and anti-mouse (Solarbio, SE131), 1:10,000.

### Yeast two-hybrid (Y2H) screening and protein interaction assays

Y2H screening was performed using a Matchmaker Gold Yeast two-hybrid system (Clontech). The truncated coding region (*BON1-A*, encoding the interacting domain) was introduced into the pGBKT7 (BD) vector and used as bait to screen a cDNA library generated from mixed samples of Arabidopsis seedlings infected or not with *Pseudomonas syringae* pv. *tomato* DC3000 (*Pst*). The coding sequence fragments encoding SERK4-KD, SERK1-KD and SERK2-KD and several additional Arabidopsis proteins including BRI1, BKI1, BIK1, TTL1, TLL4, BSK1, BSK3, and BIN2, were individually cloned into the pGADT7 (AD) vector. The gene fragments encoding ZmBON1 and the predicted ZmSERK4/BKK1 (GRMZM2G384439) kinase domain were individually cloned into the pGBKT7 and pGADT7 vector. The constructs were co-transformed in pairs into yeast strain Y2H Gold. The transformants were grown on synthetic complete (SC) medium −Leu/−Trp (Coolaber, Cat#CG5641) and SC medium −Ade/−His/−Leu/−Trp (Coolaber, Cat#PM2113) supplemented with 20 g/L glucose and X-a-Gal to assess interactions between various clone combinations.

### Split luciferase complementation (SLC) assays

The coding sequences of *BON1* and *ZmBON1* were inserted into the pCAMBIA1300-cLuc vector, and the coding sequences of *SERK4*, *SERK1*, *SERK2*, *BRI1*, *PIN1*, *ZmSERK4*, and *ZmCDPK7* were inserted into the pCAMBIA1300-nLuc vector. The resulting constructs were transformed into Agrobacterium (strain GV3101). After the bacteria were cultured in liquid LB medium to an OD at 600 nm of 1.2–1.8 at 28 °C, the cultures were pelleted by centrifugation and resuspended in infiltration buffer (10 mM MES pH 5.6, 10 mM MgCl₂, 150 μM acetosyringone) to a final OD at 600 nm of 1.0. The suspensions for infiltration were prepared by mixing the bacteria carrying the nLuc fusion, the nLuc fusion and the silencing inhibitor strain pSoup-p19 in a 1:1:1 ratio (v/v/v) and incubated at room temperature for 3 h. The bacterial suspensions were infiltrated into young *N. benthamiana* leaves. At 2 days post inoculation, the *Nicotiana* leaves were infiltrated with luciferin solution (1 mM D-fluorescein potassium salt and 0.01% [v/v] TritonX-100) and kept in the dark for 5 min, then the luciferase activity was imaged with a NightShade LB985 Plant Imaging System (Berthold Technologies).

### Co-immunoprecipitation assays

To test the protein interaction between BON1 and SERK4 and BRI1, the coding sequences of *BON1*, *SERK4*, and *BRI1* were inserted into the pSuper1300-35S-HA, pSuper1300-35S-3xFlag-mCherry, and pSuper1300-35S-GFP vectors, respectively. The resulting constructs were co-transfected in Col-0 protoplasts prepared following previously described methods[74]. Transfected protoplasts were incubated for 18 h at room temperature and used for protein extraction as described above. The supernatants were incubated with anti-HA beads (Lablead, HNM-25-1000) for 4 h at 4 °C. The collected beads were washed four times with 1× Tris-buffered saline (TBS) buffer, and the co-immunoprecipitated proteins were eluted from the beads in elution buffer (0.1 M glycine-HCl, pH 3.0).

To determine the interactions between BRI1 and SERKs in *pad4-1* and *bon1-1 bon2-2 bon3-3 pad4-1* mutants after mock and eBL treatments, total proteins from at least 50 10-day-old seedlings per treatment per genotype were extracted as described above. IgG Magnetic Beads (ROCKLAND, RLK00 1800) coupled with α-SERK3 antibodies (Agrisera, AS12 1858) and anti-FLAG Beads (Lablead, FNM-25-1000)

were used for the detection of the BRI1-SERK3 and BRI1-SERK4-FLAG interactions, respectively.

## Subcellular localization assay

The coding regions of *BON1*, *SERK3,* and *SERK4* were inserted into the pRPTL2-35S-eGFP vector. The resulting constructs were transformed into the Arabidopsis protoplasts prepared following previous described methods[74]. Transfected protoplasts were incubated for 18 h at room temperature. The subcellular localization of BON1 and SERK4 was analyzed in the Col-0 wild-type Arabidopsis protoplasts, and the subcellular localization of SERK3 and SERK4 were compared and analyzed in *pad4-1* or *bon1-1bon2-2bon3-3pad4-1* mutant protoplasts. Localization of all fusion proteins were examined under confocal microscopy (Nikon A1+).

## Phosphorylation assays

To examine the phosphorylation levels of native BRI1 and SERK3 in *pad4-1* and *bon1-1 bon2-2 bon3-3 pad4-1* after mock and 2 µM eBL treatment, total proteins were extracted from at least 50 seedling per treatment per genotype as described above and incubated with IgG Magnetic Beads (ROCKLAND, RLK00 1800) coupled with anti-BRI1 (Agrisera, AS12 1859) and anti-SERK3 (Agrisera, AS12 1858) antibodies or with anti-FLAG M2 Magnetic Beads (Sigma, M8823) for 4 h at 4 °C to respectively immunoprecipitate BRI1, SERK3, or SERK4-FLAG. Immunoprecipitated BRI1 was detected by anti-BRI1 (Agrisera, AS12 1859, 1:5000), anti-phosphoserine (anti-pSer) (Sigma, P5747, 1:500) and anti-phosphothreonine (anti-pThr) (Cell Signaling, 9381, 1:500) antibodies, respectively. Immunoprecipitated SERK3 was examined with anti-SERK3 (Agrisera, AS12 1858, 1:5000) and anti-phosphothreonine (anti-pThr) (Cell Signaling, 9381, 1:500) antibodies, respectively. To examine the phosphorylation levels of SERK4 in *p35S::SERK4-FLAG/pad4-1* and *p35S::SERK4-FLAG/bon1-1 bon2-2 bon3-3 pad4-1* seedlings after mock and 2 µM BL treatment, proteins extracted from these seedlings were mixed with anti-FLAG M2 Magnetic Beads (Sigma, M8823) for 4 h at 4 °C. Immunoprecipitated SERK4-FLAG proteins were detected by anti-FLAG (Sigma, F1804, 1:2500) and anti-phosphothreonine (anti-pThr) (Cell Signaling, 9381, 1:500) antibodies, respectively.

To examine the phosphorylation levels of BRI1 in protoplasts, the *p35S::BRI1-GFP* construct was transformed into protoplasts isolated from *pad4-1* and *bon1-1 bon2-2 bon3-3 pad4-1* mutant plants. After incubation for 16 h with mock or 10 nM eBL treatment, total proteins were extracted with in-gel buffer (50 mM Tris-HCl pH 7.5, 5 mM EDTA, 5 mM EGTA, 25 mM NaF, 1 mM $Na_3VO_4$, 20% [v/v] glycerol, 5 mM DTT, 1× protease inhibitor cocktail) and Immunoprecipitated with anti-GFP agarose (Chromotek, gta20). BRI1-GFP was detected with anti-GFP antibody (TransGen Biotech, HT801, 1:5000), and phosphorylation of BRI1 was detected with anti-pThr antibody (Cell Signaling, 9381, 1:500).

## Reporting summary

Further information on research design is available in the Nature Portfolio Reporting Summary linked to this article.

## Data availability

The transcriptomics data for maize and Arabidopsis of this article are available in the NCBI database under the accession numbers PRJNA907414 and PRJNA907415. The phosphoproteomics data of this article are available in the iProX database under the accession number IPX0005474000. The whole transcriptomic data for Col-0, *bak1-4*, *pad4-1,* and *bon1bon2bon3pad4-1* and the whole phosphoproteomic data for *pad4-1* and *bon1bon2bon3pad4*-1 under mock and eBL treatment are provided as Supplementary data files. Source data are provided with this paper.

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

## Acknowledgements

This work was supported by grants from the National Natural Science Foundation of China (grant number U2004207, to M.G.), National Key Research and Development Program of China (grant number 2022YFD1201801, to M.G.), Henan Province Major Science and Technology Project (grant number 221100110300, to J.T.), the Fund for Distinguished Young Scholars in Henan (grant number 212300410007, to M.G.), and the Zhongyuan Thousand Talents Program (grant number ZYQR201912168, to M.G.).

## Author contributions

M.G. conceived the project. M.G. and T.J. designed the experiments. T.J. performed most of the experiments. Y.W. performed the Y2H and SLC assays. M.G., J.T., D.W., S.Y. and J.H. contributed to the plant materials. Y.Y., J.L., L.X. and G.Q. contributed to the propagation and phenotyping of plant materials. X.M. contributed to the transcriptomics analysis. X.W. performed the BRI1 phosphorylation assay in protoplasts. All authors analyzed the data. T.J. and M.G. wrote the manuscript with comments from all authors.

## Competing interests

The authors declare no competing interests.
