## [Peer Review File · Nature Communications]

REVIEWER COMMENTS

Reviewer #1 (Remarks to the Author):

This manuscript described the regulation of BR signaling by BON proteins. Previous studies in Arabidopsis and rice demonstrated that BON proteins are critical for plant growth, development, defense, and stress responses. These plasma membrane-associated copine proteins interact with RLKs, calcium pumps, and other C2 domain proteins. However, the most conserved roles for BON proteins still unclear. Through genetic and chemical studies, the authors found that BON proteins are critical for BR signaling in maize and Arabidopsis. They found that BON proteins interact with SERK kinases, and control BR signaling through regulating BRI1-SERK interaction and transphosphorylation. The manuscript is well written and the finding has been well demonstrated and discussed. Overall, this is an interesting story that exhibited new functions of BON proteins in maize and Arabidopsis. I have the following questions.

1, The authors showed that mutation of ZmBON1 leads to dwarfism but no autoimmunity. I noticed that the analyses of plant growth, PR gene expression and SA measurement were performed in different developmental ages or stages. However, the dwarfism phenotype of bon1 mutant alters according to environmental changes or developmental stages. My question is that does zmbon1 mutants have dwarfism phenotype at the exact same conditions for gene expression and SA analyses.

2, Similar like maize, rice genome also lacks TNLs. However, previous studies demonstrated that OsBon1 and OsBon3 are suppressors of broad spectrum disease resistance. Thus, whether ZmBONs also function in immunity should be carefully analyzed.

3, In figure 5b, the Co-IP assays lack an appropriate negative control: BKK1-Flag with another HA tagged protein. In figure 5c, the negative controls have no signals. This raise a question whether the negative n/c-LUC control proteins are expressed. It must be noticed that some n/c-LUC con This manuscript described the regulation of BR signaling by BON proteins. Previous studies in Arabidopsis and rice demonstrated that BON proteins are critical for plant growth, development, defense, and stress responses. These plasma membrane-associated copine proteins interact with RLKs, calcium pumps, and other C2 domain proteins. However, the most conserved roles for BON proteins are still unclear. Through genetic and chemical studies, the authors found that BON proteins are critical for BR signaling in maize and Arabidopsis. They discovered that BON proteins interact with SERK kinases and control BR signaling by regulating BRI1-SERK interaction and transphosphorylation. The manuscript is well written, and the finding has been well demonstrated and discussed. Overall, this exciting story exhibited new functions of BON proteins in maize and Arabidopsis. I have the following questions.

1, The authors showed that mutation of ZmBON1 leads to dwarfism but no autoimmunity. I noticed that plant growth, PR gene expression, and SA measurement analyses were performed in different ages or developmental stages. However, the dwarfism phenotype of the bon1 mutant alters according to environmental changes or developmental stages. Do zmbon1 mutants have dwarfism phenotype at the same conditions for gene expression and SA analyses?

Similar to maize, the rice genome also lacks TNLs. However, previous studies demonstrated that OsBon1 and OsBon3 are suppressors of broad-spectrum disease resistance. Thus, whether ZmBONs also function in immunity should be carefully analyzed.

3, In figure 5b, the Co-IP assays lack an appropriate negative control: BKK1-Flag with another HA-tagged protein. In figure 5c, the negative controls have no signals. This raises the question of whether the negative n/c-LUC control proteins are expressed. It must be noticed that some n/c-LUC constructs have no start codon, and thus the proteins could not be translated without fused genes. I suggest the authors use other n/c-LUC fused proteins as negative controls.

4, In the immunoprecipitated protein kinase assays shown in Figure 6, the authors used the same antibodies for immunoprecipitation and western blot. Did the authors notice strong IgG signals during the western blot? Moreover, protein kinase assays with ATP-gamma-S, or P32-labelled ATP, can illustrate the BON1-mediated BRI1-SERK transphosphorylation more clearly.

Reviewer #2 (Remarks to the Author):

This work has demonstrated that BON gene family influences the plant growth. These proteins might interact with each member of BRI-SERK receptor complex, and they are necessary for the interaction between receptor complex subunits and the protein phosphorylation of these two components of BR receptor complexes. Phosphoproteomics was performed on two plant mutant lines, pad4 and pad4bon1bon2bon3, which were separately treated with hormone. These phosphoproteomics results have shown some putative phosphorylation sites existing on both BRI1 and SERK1 proteins without replication data presented. Quantitative data have only one ratio of PTM peptides between two samples in supplemental table.

Major modifications of paper are seriously needed to present a nice and credible story to readers:

1) Re-write the Introduction to emphasize on the BR signalling and its effect on growth of under both and dark conditions and adult stages. Remove the part of the possible effects of Bon family on the autoimmunity and lesion phenotype.

2) Change all labels of BKK1 and BAK1 to SERK4 and SERK3, respectively, in all figures, abstract, title, legends and results. Mark both names in Materials and methods.

3) Mention the plant species used for the experiment in each figure legend and method describing the co-immunoprecipitation immunoblot assay, split LUC assay and yeast two hybrid as well as phosphoproteomic experiments and morphological assays. Currently, authors included three plant species, maize, Arabidopsis and tobacco, as well as different types of tissues or cell lines to perform different biochemical and molecular experiments, it is very confusing. Can we assume that BRI-SERK complexes, kinase activities and phosphorylation levels as well as functions to the downstream signalling components of these subunits are identical in all protoplasts, dark and light growing seedlings and adult plants in all three plant species ?

It is very hard to draw conclusions based separately performed results from multiple possible signalling systems (the 3 types of plant materials x 3 species). We have to assume that all these signalling systems share a common BL signalling pathway. If we take into account of dark and light growth conditions, we may get more different signalling systems. Authors should justify why you think that these BL signalling pathways are identical under these different plant systems in the Introduction so that those combinatorial data collected from all types tissues and plant species, and even all three different individual members of SERKs, should demonstrate a uniform BR signalling pathway model in three plant species. That is tough but doable.

4) Anti-pSer or anti-Thr will not tell which p-site(s) on BRI and SERK is (are) phosphorylated on SERK or BRI proteins. Phosphoproteomics data are rudimentary. The manuscript and supplementary tables neither showed the 3 replicates nor quantitative data of multiple hypothesis corrections on t-test or z-test values nor amount of tissues and protein samples used for quantitative phosphoproteomics.

In short, authors should learn how to present a quantitative phosphoproteomics data in a paper. In Materials and Methods, there is no detailed tissue collection data, protein and peptide preparation data nor chromatography data and MS specification. Authors are strongly recommended to show data and results, at least in supplementary figure and tables, on details of all steps of phosphoproteomics.

Anti-phosphosite antibodies should be made to show which phosphosites of BRI and SERK are phosphorylated upon BL induction to conform and validate the phosphoproteomics data. If so, author may only need to present 3 replicates of phosphoproteomic data rather than a statistics-based

quantitative phosphoproteomics results (assuming that authors may find quantitative phosphoproteomics is hard to achieve).

Current pulldown immunoblot assay data only showed the necessity of BON proteins for interaction intensity between BRI and SERK members and phosphorylation on both proteins. But there is no evidence to show how BON family proteins participate in activating the kinase activity of both subunits of BRI-SERK protein complex toward each other. The current model should be modified to show three members of BON-BRI-SERK complex in a triangle conformation based on pulldown and split LUC assay results. I am wondering why split LUC assay was not performed on proteins of BRI and SERK under both *bon1bon2bon3 pad4* and *pad4* mutant backgrounds???

Why did not phosphoproteomics find phosphorylation site(s) of BON proteins if they are in contact with two families of kinases of BRI and SERKs?? Does Bon protein bind the phosphorylated BRI and SERK proteins to prevent dephosphorylation or dephosphorylated form to activate kinase activity? Does dephosphorylation dissociate BRI and SERK? Would it be possible that both BRI and SERK autophosphorylate themselves and the phosphosrylated isoforms interact with each other or with BON proteins???

The current story is vague. At least, maybe authors should make p-site specific antibodies against BRI1 and SERKs phosphosites found from mutant lines and perform the immunoblots to show how these sites' phosphorylation change upon BL induction under two mutant backgrounds.

Reviewer #3 (Remarks to the Author):

Copine proteins are a group of broadly presented proteins in eukaryotes, including plants and animals. BONZAI (BON) proteins are plasma membrane associated copine proteins found in plants. They were initially identified as negative regulators in plant defense responses, because *bon1* (also named *cpn1*) mutants show autoimmune phenotypes, including cell death and seedling lethality. Further analyses indicated that the autoimmune phenotypes rely on the presence of TNL-type R genes. In this manuscript, the authors studied the functions of BON homologous proteins in maize (no TNL-type R genes in maize) and found a *Zmbon1* mutant does not show autoimmune phenotypes but exhibits a dwarfed phenotype similar to the defective phenotype of BRI1, the receptor of brassinosteroids (BRs). The authors did a number of experiments, including a root inhibition analysis and the downstream gene expression in response to eBL treatment and confirmed that the BR signaling pathway in *Zmbon1* is indeed impaired. The authors also generated a *bon1-1 bon2-2 bon3-3 pad4-1* quadruple mutant in which all three BON genes in Arabidopsis are knocked out and the autoimmune response pathway is blocked. The quadruple mutant again showed dwarfed phenotype similar to a weak BRI1 mutant phenotype.

Overall, I find the research is interesting. But I do have a number of suggestions and concerns listed here. I hope the authors will find them useful to improve this manuscript.

1) The major concern is that the detailed molecular mechanism of BON1 in regulating BR signaling is lacking in the current version of the manuscript. The authors should analyze whether the interactions between BON1 and BAK1 or BRI1 is due to the alteration of protein stability (for example, the abundance of BRI1 and BAK1), or protein trafficking during the maturation of two RLKs in the endomembrane system, or binding affinity between BRI1 and BAK1, etc. To publish a paper in a good journal like Nature Communications, a step forward analysis is needed.

2) I am wondering why the authors did not use WS accession because it lacks TNL-type R genes. The triple mutant should be easily generated by using a gene editing approach.

3) Most of the analyses were conducted in the quadruple mutant *bon1-1 bon2-2 bon3-3 pad4-1*. Complementation analyses are required to further solidify the conclusion. The authors should transform BON1, BON2, and BON3 back to the quadruple mutant (driven by their own promoters) and test whether the morphology and responsive gene expression levels are recovered.

4) I noticed the authors tested a number of BR responsive genes. To see the BR signal output, most people use the response of SAUR-AC1. The authors should add the expression levels of SAUR-AC1 in their tests.

5) Figure 2d, the grey and orange colors sometimes are hardly to tell apart due to the existence of red dots. It's better to change the colors.

6) Figure 3a, the triple mutant not only regulates autoimmunity but also regulates plant development. The up part, therefore, is not complete.

7) Figure 3c, eBL should be marked at the X axis.

8) To confirm the specificity of the antibodies to against BRI1, BZR1, etc, negative controls should be used. Unfortunately, none of the negative controls were included in the manuscript. For example, an anti-BRI1 antibody should not have a cross reaction band in a *bri1* mutant background. Likewise, an anti-BZR1 antibody should not have a band in a *bzr1* mutant background.

9) Figure 3c, Y axis should start from 0, not 10. It is wrong to use such a way to exaggerate the difference.

10) The authors showed the interaction between BON1 and BRI1 only in a BIFC analysis. They should confirm it by using additional approaches.

Response to reviewers

Dear reviewers,

Thank you for the comments and suggestions on the improvement of our manuscript. We have revised the manuscript comprehensively according to all of your comments. Please see the revised manuscript and the following response to reviewers for details, and let us know if you may have additional questions or concerns. We appreciate your feedback a lot.

Best regards.

REVIEWER COMMENTS

Reviewer #1 (Remarks to the Author):

This manuscript described the regulation of BR signaling by BON proteins. Previous studies in Arabidopsis and rice demonstrated that BON proteins are critical for plant growth, development, defense, and stress responses. These plasma membrane-associated copine proteins interact with RLKs, calcium pumps, and other C2 domain proteins. However, the most conserved roles for BON proteins still unclear. Through genetic and chemical studies, the authors found that BON proteins are critical for BR signaling in maize and Arabidopsis. They found that BON proteins interact with SERK kinases, and control BR signaling through regulating BRI1-SERK interaction and transphosphorylation. The manuscript is well written and the finding has been well demonstrated and discussed. Overall, this is an interesting story that exhibited new functions of BON proteins in maize and Arabidopsis. I have the following questions.

1, The authors showed that mutation of ZmBON1 leads to dwarfism but no autoimmunity. I noticed that the analyses of plant growth, PR gene expression and SA measurement were performed in different developmental ages or stages. However, the dwarfism phenotype of bon1 mutant alters according to environmental changes or developmental stages. My question is that does zmbon1 mutants have dwarfism phenotype at the exact same conditions for gene expression and SA analyses.

Response: Thanks for the comments.

Yes. We used 4-leaf-age seedlings grown in growth chamber at 22°C for *PR* gene expression analysis and SA measurement. The plants did show obvious dwarf phenotype (Extended Data Fig.6a). We also observed dwarf phenotype for the 2-leaf-age seedlings grown at the same condition (Extended Data Fig.6c). We have performed additional analysis of *PR* gene expression using newly collected plants at the two stages. There are no differences of *ZmPR1* and *ZmPR5* expression between KN5585 WT and *Zmbon1* mutant at both stages (Extended Data Fig.6b, d).

2, Similar like maize, rice genome also lacks TNLs. However, previous studies demonstrated that OsBon1 and OsBon3 are suppressors of broad spectrum disease resistance. Thus, whether ZmBONs also function in immunity should be carefully analyzed.

Response: Thanks for the comments.

To obtain plants with activated immunity by gene editing of *ZmBON1* and *ZmBON3* in maize is actually one of our original purposes. However, after careful examination of defense gene expression by transcriptomics analysis and qRT-PCR, and hormone level of the mutant plants at different stages, we didn't see activated defense gene expression and changes of defense-related hormones (Figure.1, Extended Data.Fig.5 and Extended Data Fig.6). In contrast, *les30*, an autoimmune mutant we studied at the same time, showed obvious characteristics for defense activation, constitutive lesion formation, activated *PR* gene expression and SA accumulation (Li et al., 2022, Response Fig. 1). We also investigated the disease symptoms of plants grown at two different locations (Zhengzhou and Sanya, China) for 3 years. We didn't observe enhanced resistance of *Zmbon1* and *Zmbon3* mutants to any diseases at any growth stages. Because of these observations, we started to consider if *ZmBON1* and *ZmBON3* may have other roles, which led to current discovery of BON's function in BR signaling.

Response Fig.1 | Characteristics of activated defense response in a typical maize autoimmune mutant *les30*.

3, In figure 5b, the Co-IP assays lack an appropriate negative control: BKK1-Flag with another HA-tagged protein. In figure 5c, the negative controls have no signals. This raises the question of whether the negative n/c-LUC control proteins are expressed. It must be noticed that some n/c-LUC constructs have no start codon, and thus the proteins could not be translated without fused genes. I suggest the authors use other n/c-LUC fused proteins as negative controls.

Response: Thanks for the suggestions.

We agree with the concern and have re-performed the Co-IP and SLC assay respectively using more suitable negative controls. Now, the plasma-membrane localized protein PIN1 and ZmCDPK7 were used as negative controls for AtBON1 and ZmBON1 in Arabidopsis and maize, respectively. For Co-IP assay, BKK1/SERK4-FLAG was co-immunoprecipitated with BON1-HA but not with PIN1-HA (Fig.5b), For SLC assay, While LUC signal was observed in leaves co-infiltrated with SERK4-nLuc and cLuc-BON1 (Fig.5c), SERK1-nLuc and cLuc-BON1 (Extended Data Fig.11a), SERK2-nLuc and cLuc-BON1 (Extended Data Fig.11b), and ZmSERK4-nLuc and cLuc-ZmBON1 (Fig.5e), no LUC signal was observed in leaves co-infiltrated with PIN1-nLUC and cLuc-BON1 (Fig.5c and Extended Data Fig.11), or ZmCDPK7-nLuc and cLuc-ZmBON1 (Fig.5e). These data indicate that BON proteins interact specifically with SERK proteins in Arabidopsis and maize.

4, In the immunoprecipitated protein kinase assays shown in Figure 6, the authors used the same antibodies for immunoprecipitation and western blot. Did the authors notice strong IgG signals during the western blot? Moreover, protein kinase assays with ATP-gamma-S, or P32-labelled ATP, can illustrate the BON1-mediated BRI1-SERK transphosphorylation more clearly.

Response: Thanks you for the comments and suggestions.

Yes, we did observe strong IgG signals during the western blot (Response Fig. 2). Fortunately, the size of BRI1 (>130 kD) and BAK1 (>70 kD) is much bigger than the IgG bands, therefore, we could clearly observe the BRI1 and BAK1 bands. According to previous studies (Hua et al., 2001; Li et al., 2010; Wang et al., 2020), BON1 didn't appear to have enzyme activity. It may instead act as a structural protein promoting the BRI1 and SERK interaction, thereby enhancing their reciprocal transphosphorylation. We did consider performing in vitro kinase assay using purified BON1, BRI1 and SERK proteins. However, since all the three proteins are membrane proteins, they were not easily purified, and their interaction and functioning are largely dependent on their plasma membrane localization, in vitro assay of three proteins using a non-membrane system may not work properly.

Thanks again for all the comments and we wish your concerns have been addressed.

Response Fig.2 |Full images of the immunoblots for immunoprecipitated protein kinase assay of BRI1 (a) and SERK3 (b)

Reference:

- Hua, J., Grisafi, P., Cheng, S.H., and Fink, G.R. (2001).** Plant growth homeostasis is controlled by the Arabidopsis BON1 and BAP1 genes. *Genes Dev* **15**, 2263-2272.
- Li, Y., Gou, M., Sun, Q., and Hua, J. (2010).** Requirement of calcium binding, myristoylation, and protein-protein interaction for the Copine BON1 function in Arabidopsis. *J Biol Chem* **285**, 29884-29891.
- Li, J., Chen, M., Fan, T., Mu, X., Gao, J., Wang, Y., Jing, T., Shi, C., Niu, H., Zhen, S., Fu, J., Zheng, J., Wang, G., Tang, J., and Gou, M. (2022).** Underlying mechanism of accelerated cell death and multiple disease resistance in a maize lethal leaf spot 1 allele. *J Exp Bot* **73**, 3991-4007.
- Wang, Q., Jiang, M., Isupov, M.N., Chen, Y., Littlechild, J.A., Sun, L., Wu, X., Wang, Q., Yang, W., Chen, L., Li, Q., and Wu, Y. (2020).** The crystal structure of Arabidopsis BON1 provides insights into the copine protein family. *Plant J* **103**, 1215-1232.

Reviewer #2 (Remarks to the Author):

This work has demonstrated that BON gene family influences the plant growth. These proteins might interact with each member of BRI-SERK receptor complex, and they are necessary for the interaction between receptor complex subunits and the protein phosphorylation of these two components of BR receptor complexes.

Phosphoproteomics was performed on two plant mutant lines, *pad4* and *pad4bon1bon2bon3*, which were separately treated with hormone. These phosphoproteomics results have shown some putative phosphorylation sites existing on both BRI1 and SERK1 proteins without replication data presented. Quantitative data have only one ratio of PTM peptides between two samples in supplemental table.

Major modifications of paper are seriously needed to present a nice and credible story to readers:

1) Re-write the Introduction to emphasize on the BR signalling and its effect on growth of under both and dark conditions and adult stages. Remove the part of the possible effects of Bon family on the autoimmunity and lesion phenotype.

Response: Thank you for the comments and suggestions.

We have rewritten the introduction to emphasize the BR signaling (Page 2-3, line 41-75). We agree that the introduction of autoimmunity and lesion phenotype of BON family in Arabidopsis is a little tedious. Since we used the *bon1-1 bon2-2 bon3-3 pad4-1* quadruple mutant that is depleted with autoimmunity, to avoid confusion about the materials and concepts of previous studies, we kept the introduction about the autoimmunity and lesion phenotype related to BON1, however, we have reorganized this part to be more concise (Page 3, line 86-102).

2) Change all labels of BKK1 and BAK1 to SERK4 and SERK3, respectively, in all figures, abstract, title, legends and results. Mark both names in Materials and methods.

Response: Thanks for the kind suggestions.

We have changed all the labels of BKK1 and BAK1 to SERK4 and SERK3 throughout the manuscript. Both names were also marked in Materials and Methods.

3) Mention the plant species used for the experiment in each figure legend and method describing the co-immunoprecipitation immunoblot assay, split LUC assay and yeast two hybrid as well as phosphoproteomic experiments and morphological assays. Currently, authors included three plant species, maize, Arabidopsis and tobacco, as well as different types of tissues or cell lines to perform different biochemical and molecular experiments, it is very confusing. Can we assume that BRI-SERK complexes, kinase activities and phosphorylation levels as well as

functions to the downstream signalling components of these subunits are identical in all protoplasts, dark and light growing seedlings and adult plants in all three plant species?

It is very hard to draw conclusions based separately performed results from multiple possible signalling systems (the 3 types of plant materials x 3 species). We have to assume that all these signalling systems share a common BL signalling pathway. If we take into account of dark and light growth conditions, we may get more different signalling systems. Authors should justify why you think that these BL signalling pathways are identical under these different plant systems in the Introduction so that those combinatorial data collected from all types tissues and plant species, and even all three different individual members of SERKs, should demonstrate a uniform BR signalling pathway model in three plant species. That is tough but doable.

Response: Thanks for the comments and suggestions.

Sorry for the confusion. We have mentioned the plant species used for the experiments in all figure legends and methods. We have also introduced the conserved classical components of BR signaling in different plants in the Introduction (Page 3, line 68-75). According to previous studies, the core components in BR signaling, including BAK1/SERKs and BRI1, are highly conserved in monocot and dicot (Response Fig.3) (Nakamura et al., 2006; Bai et al., 2007; Koh et al., 2007; Li et al., 2009; Kir et al., 2015; Kim and Russinova, 2020). Likewise, BON protein sequences are also highly conserved in eukaryotes especially in plants (Creutz et al., 1998; Zou et al., 2016). Therefore, it is reasonable that BON proteins work with BAK1/SERKs and BRI1 to regulate BR signaling in a conservative manner in monocot and dicot. In this study, we first discovered the function of *ZmBON1* in BR signaling due to the absence of interference by autoimmunity in maize that does not have TNL-type R proteins. With this clue, the conserved function of BON1 in BR signaling was further confirmed using the *bon1-1 bon2-2 bon3-3 pad4-1* quadruple mutant depleted with autoimmunity in Arabidopsis. Besides, both BON1 and *ZmBON1* interact with SERK proteins, confirming the conservative role of BON proteins in BR signaling in maize and Arabidopsis. Since *N. benthamiana* was used as a well-established transient expression system for Co-IP and SLC assay in many studies, we used it to confirm the interaction between BON and the BR receptor complex.

Besides, our data support that BON proteins regulate BR signaling under both light and dark conditions. For instance, hypocotyl elongation in dark was known to be controlled by BR, and was used as a typical phenotype to identify BR signaling mutants in Arabidopsis (Fridman and Savaldi-Goldstein, 2013; Tang et al., 2008; Tang et al., 2011; Imkampe et al., 2017; Zhu et al., 2017; Amorim-Silva et al., 2019). In our study, while the hypocotyls of wild type plants elongated normally in the dark, that of *bon1-1 bon2-2 bon3-3 pad4-1* plants was largely attenuated (Figure 3d, e).

In sum, we used different plant systems to validate the conserved function of BON proteins, and these data support the conclusion that BON proteins' function in BR

signaling is conserved. To better fit our data, we changed the manuscript title to “Copine proteins are required for brassinosteroid signaling in maize and Arabidopsis”

Response Fig.3 | Evolution of BR signaling (Kim & Russinova, 2020)

4) Anti-pSer or anti-Thr will not tell which p-site(s) on BRI and SERK is (are) phosphorylated on SERK or BRI proteins. Phosphoproteomics data are rudimentary. The manuscript and supplementary tables neither showed the 3 replicates nor quantitative data of multiple hypothesis corrections on t-test or z-test values nor amount of tissues and protein samples used for quantitative phosphoproteomics.

In short, authors should learn how to present a quantitative phosphoproteomics data in a paper. In Materials and Methods, there is no detailed tissue collection data, protein and peptide preparation data nor chromatography data and MS specification. Authors are strongly recommended to show data and results, at least in supplementary figure and tables, on details of all steps of phosphoproteomics.

Anti-phosphosite antibodies should be made to show which phosphosites of BRI and SERK are phosphorylated upon BL induction to conform and validate the phosphoproteomics data. If so, author may only need to present 3 replicates of phosphoproteomic data rather than a statistics-based quantitative phosphoproteomics results (assuming that authors may find quantitative phosphoproteomics is hard to achieve).

Response: Thanks for the comments and suggestions.

We have added more details in the Materials and Methods, including protein extraction and digestion, phosphopeptides enrichment, LC-MS/MS analysis and identification and quantitation of phosphorylated proteins (Extended Data. File 1, Page 18, line 489-492 and line 500-501). We have also performed more biological

replicates of immunoblots and the new data have been presented (**Extended Data Fig.17**), which further confirmed our previous results.

We originally planned to perform 3 replicates of phosphoproteomics to reveal the general changes of phosphorylation events. However, according to our discussion with the phosphoproteomics staff, we realize that phosphoproteomics data many not be repeated well in different replicates due to technical limits and the experiments are costly. We thus preferred to perform one replicate of phosphoproteomics, and then validate the detected changes of phosphorylation by immunoblots.

Our phosphoproteomics data implied that the phosphorylation cascade, represented by BRI1, SERKs, BKI1, BSKs, and BSLs, was activated by BR treatment in the wild type, and such activation was attenuated in the *bon1-1 bon2-2 bon3-3 pad4-1* mutant (**Extended Data Fig.16**). Specially, the detected phosphorylation site (S1166) of BRI1 is a proved site of transphosphorylation by co-receptor SERKs (**Wang et al., 2008**). To confirm that the transphosphorylation of BRI1 is impaired in *bon1-1 bon2-2 bon3-3 pad4-1*, we did try to generate the antibodies for the phosphorylated pBRI1^{S1166}. Unfortunately, we have not yet got the specific pBRI1^{S1166} antibodies despite several attempts, and no such antibodies were used in previous publications. Instead, we immunoprecipitated BRI1 and SERKs, and detected their overall phosphorylation status using anti-pThr or anti- pSer antibodies. To double confirm the phosphorylation changes, we used both 10-day-old seedlings (1 biological replicate) and protoplasts transiently expressing the BRI1-GFP (4 biological replicates) for such assay (**Fig.6 and Extended Data Fig.18**). **To further confirm our data, we have presented two additional biological replicates of experiments using 10-day-old seedlings (Extended Data Fig.17)**. These data again confirmed our previous results.

Since BON proteins do not seem to have kinase activities according to our previous studies (**Hua et al., 2001; Li et al., 2010**) as well as the analysis of BON's crystal structure (**Wang et al., 2020**), it is unlikely that BON proteins could directly phosphorylate SERKs and BRI1. Instead, our results support that BON proteins act as structural proteins, which promote the interaction between BRI1 and SERKs, thus generally affecting the reciprocal transphosphorylation between SERKs and BRI1. **Therefore, the detection of overall phosphorylation level of BRI1 and SERKs is actually more important than detection of the specific phosphorylation sites in this study.**

In sum, both phosphoproteomics data and immunoblot experiments indicate that the general phosphorylation of BRI1 and SERKs are attenuated when BON genes are fully knocked-out. However, if there is strict requirement of replicates of phosphoproteomics to be used as preliminary data for the journal, we could delete the phosphoproteomics data because the immunoblots of BRI1 and SERK protein using the Anti-pSer or anti-Thr antibodies are confirmed by totally 3 and 4 biological replicates in plants and protoplasts. Those data provide sound evidence to draw the conclusion that BON proteins are required for the general transphosphorylation between BRI1 and SERK proteins.

Current pulldown immunoblot assay data only showed the necessity of BON proteins for interaction intensity between BRI and SERK members and phosphorylation on both proteins. But there is no evidence to show how BON family proteins participate in activating the kinase activity of both subunits of BRI-SERK protein complex toward each other. The current model should be modified to show three members of BON-BRI-SERK complex in a triangle conformation based on pulldown and split LUC assay results. I am wondering why split LUC assay was not performed on proteins of BRI and SERK under both *bon1 bon2 bon3 pad4* and *pad4* mutant backgrounds???

Why did not phosphoproteomics find phosphorylation site(s) of BON proteins if they are in contact with two families of kinases of BRI and SERKs??

Does Bon protein bind the phosphorylated BRI and SERK proteins to prevent dephosphorylation or dephosphorylated form to activate kinase activity?

Does dephosphorylation dissociate BRI and SERK?

Would it be possible that both BRI and SERK autophosphorylate themselves and the phosphorylated isoforms interact with each other or with BON proteins???

The current story is vague.

At least, maybe authors should make p-site specific antibodies against BRI1 and SERKs phosphosites found from mutant lines and perform the immunoblots to show how these sites' phosphorylation change upon BL induction under two mutant backgrounds.

Response: Thanks for the comments and suggestions.

BON proteins were originally named copine, namely partner or chaperone. Based on previous studies, BON proteins did not seem to have enzyme activity. Therefore, we suspect that BON proteins act as chaperones or structural proteins mediating BRI1 and SERK interaction. According to the yeast two-hybrid experiment, which reveals the direct protein-protein interaction, BON proteins appear to directly interact with SERK proteins but not the BRI1 protein (Fig.5a and Extended Data Fig.12), whereas, the Co-IP and SLC assay support that BON proteins associate with BRI1 protein likely through its direct interaction with SERKs (Fig.5 and Fig.6). **Therefore, we propose the existence of a BON-SERK-BRI1 complex, in which SERKs interact with BONs and BRI1 directly, but BONs and BRI1 associate via SERK proteins. The complex is thus unlikely in a triangle conformation.** We have included such information in Discussion (Page 14, line 389-391).

Since SLC assay was usually performed in a transient expression system in *N. benthamiana* (or tobacco) to confirm the protein-protein interaction, and transient expression system is often less stable in Arabidopsis, we didn't use this technique to test the intensity of interaction between BRI1 and SERKs in Arabidopsis. Instead, we did Co-IP assay to compare the intensity of interaction between endogenous BRI1 and SERK proteins under both *bon1-1 bon2-2 bon3-3 pad4-1* and *pad4-1* mutant backgrounds (Fig.6a, b), and our data clearly showed that the interaction between BRI1 and SERKs are attenuated when BON proteins are fully knocked-out.

We are not sure about why phosphorylation site(s) of BON proteins are not found by phosphoproteomics. However, our previous study also indicated that not all peptides of BON1 were detectable (Li et al., 2010). Besides, unique S/T phospho-peptide was not found for the BON1 protein in public proteomics data, it is thus possible that BON1 is not usually phosphorylated in vivo. But other possibility can't be excluded.

We could not exclude the possibility that dephosphorylation and/or autophosphorylation of BRI1 or SERKs are regulated by BON proteins based on current data. We have now added this information in Discussion (Page 15, line 410-412).

We understand that you may still have concerns on the exact function of BONs in regulating the BR receptor complex formation and the subsequent phosphorylation events. We have now performed two additional experiments, the protein stability assay and the protein trafficking assay. Our data indicate that BON proteins do not appear to be required for the stability of BRI1 and SERKs (Extended Data Fig.14, Page 10-11, line 276-283), nor for the protein trafficking of the receptor complex (Extended Data Fig.15, Page 11, line 284-298). For more details, please see our response to reviewer 3. These data again support our hypothesis that BON proteins act as a chaperone or structural proteins mediating BRI1 and SERK interaction and their reciprocal phosphorylation (Figure 7). We have added the hypothesis in Discussion (Page 14, line 389-394).

Besides, according to previous studies, BON1 was known to be associated with the detergent-resistant microdomains and the PM-localization is essential for BON's function (Li et al., 2010). Interestingly, recent studies indicate the existence of preassembled BRI1-SERKs complexes in PM microdomains (Bucherl et al., 2017; Hutten et al., 2017). Therefore, it is also possible that BONs act as a structural protein affecting the PM microdomain formation or attracting the SERKs proteins transport to PM microdomain, thereby affecting BRI1-SERKs complex formation. We also believe structural analysis of BON-SERK-BRI1 complex will give us more direct answer in the future. We have included above information in Discussion (Page15, line 403-412 and line 419-422).

In addition, to better fit our data, **we have changed the manuscript title to “Copine proteins are required for brassinosteroid signaling in maize and Arabidopsis”**.

In sum, although a few questions remain to be further explained, the current study indicate that copine proteins are critical component of BR signaling in maize and Arabidopsis, which reshaped our previous understanding of BR signaling and the function of copine proteins. The discovery of BONs' conserved function in both dicot maize and monocot Arabidopsis opens a way to study steroid hormone signaling and copine proteins in a broader perspective. Based on this study, deeper understanding of BR signaling especially the mechanism for fine-tuning of BR receptor complex is expected when more comprehensive studies are performed in the future.

Thanks again for your comments and we wish the above revisions and explanations have addressed your concerns.

Reference:

- Amorim-Silva, V., Garcia-Moreno, A., Castillo, A.G., Lakhssassi, N., Esteban Del Valle, A., Perez-Sancho, J., Li, Y., Pose, D., Perez-Rodriguez, J., Lin, J., Valpuesta, V., Borsani, O., Zipfel, C., Macho, A.P., and Botella, M.A.** (2019). TTL Proteins Scaffold Brassinosteroid Signaling Components at the Plasma Membrane to Optimize Signal Transduction in Arabidopsis. *Plant Cell* **31**, 1807-1828.
- Bai, M.Y., Zhang, L.Y., Gampala, S.S., Zhu, S.W., Song, W.Y., Chong, K., and Wang, Z.Y.** (2007). Functions of OsBZR1 and 14-3-3 proteins in brassinosteroid signaling in rice. *Proc Natl Acad Sci U S A* **104**, 13839-13844.
- Bucherl, C.A., Jarsch, I.K., Schudoma, C., Segonzac, C., Mbengue, M., Robatzek, S., MacLean, D., Ott, T., and Zipfel, C.** (2017). Plant immune and growth receptors share common signalling components but localise to distinct plasma membrane nanodomains. *Elife* **6**.
- Creutz, C.E., Tomsig, J.L., Snyder, S.L., Gautier, M.C., Skouri, F., Beisson, J., and Cohen, J.** (1998). The copines, a novel class of C2 domain-containing, calcium-dependent, phospholipid-binding proteins conserved from Paramecium to humans. *J Biol Chem* **273**, 1393-1402.
- Fridman, Y., and Savaldi-Goldstein, S.** (2013). Brassinosteroids in growth control: how, when and where. *Plant Sci* **209**, 24-31.
- Hua, J., Grisafi, P., Cheng, S.H., and Fink, G.R.** (2001). Plant growth homeostasis is controlled by the Arabidopsis BON1 and BAP1 genes. *Genes Dev* **15**, 2263-2272.
- Hutten, S.J., Hamers, D.S., Aan den Toorn, M., van Esse, W., Nolles, A., Bucherl, C.A., de Vries, S.C., Hohlbein, J., and Borst, J.W.** (2017). Visualization of BRI1 and SERK3/BAK1 Nanoclusters in Arabidopsis Roots. *PLoS One* **12**, e0169905.
- Imkampe, J., Halter, T., Huang, S., Schulze, S., Mazzotta, S., Schmidt, N., Manstretta, R., Postel, S., Wierzba, M., Yang, Y., van Dongen, W., Stahl, M., Zipfel, C., Goshe, M.B., Clouse, S., de Vries, S.C., Tax, F., Wang, X., and Kemmerling, B.** (2017). The Arabidopsis Leucine-Rich Repeat Receptor Kinase BIR3 Negatively Regulates BAK1 Receptor Complex Formation and Stabilizes BAK1. *Plant Cell* **29**, 2285-2303.
- Kim, E.J., and Russinova, E.** (2020). Brassinosteroid signalling. *Curr Biol* **30**, R294-R298.
- Kir, G., Ye, H., Nelissen, H., Neelakandan, A.K., Kusnandar, A.S., Luo, A., Inze, D., Sylvester, A.W., Yin, Y., and Becraft, P.W.** (2015). RNA Interference Knockdown of BRASSINOSTEROID INSENSITIVE1 in Maize Reveals Novel Functions for Brassinosteroid Signaling in Controlling Plant Architecture. *Plant Physiol* **169**, 826-839.
- Koh, S., Lee, S.C., Kim, M.K., Koh, J.H., Lee, S., An, G., Choe, S., and Kim, S.R.**

- (2007). T-DNA tagged knockout mutation of rice OsGSK1, an orthologue of Arabidopsis BIN2, with enhanced tolerance to various abiotic stresses. *Plant Mol Biol* **65**, 453-466.
- Li, D., Wang, L., Wang, M., Xu, Y.Y., Luo, W., Liu, Y.J., Xu, Z.H., Li, J., and Chong, K.** (2009). Engineering OsBAK1 gene as a molecular tool to improve rice architecture for high yield. *Plant Biotechnol J* **7**, 791-806.
- Li, Y., Gou, M., Sun, Q., and Hua, J.** (2010). Requirement of calcium binding, myristoylation, and protein-protein interaction for the Copine BON1 function in Arabidopsis. *J Biol Chem* **285**, 29884-29891.
- Nakamura, A., Fujioka, S., Sunohara, H., Kamiya, N., Hong, Z., Inukai, Y., Miura, K., Takatsuto, S., Yoshida, S., Ueguchi-Tanaka, M., Hasegawa, Y., Kitano, H., and Matsuoka, M.** (2006). The role of OsBRI1 and its homologous genes, OsBRL1 and OsBRL3, in rice. *Plant Physiol* **140**, 580-590.
- Tang, W., Kim, T.W., Oses-Prieto, J.A., Sun, Y., Deng, Z., Zhu, S., Wang, R., Burlingame, A.L., and Wang, Z.Y.** (2008). BSKs mediate signal transduction from the receptor kinase BRI1 in Arabidopsis. *Science* **321**, 557-560.
- Tang, W., Yuan, M., Wang, R., Yang, Y., Wang, C., Oses-Prieto, J.A., Kim, T.W., Zhou, H.W., Deng, Z., Gampala, S.S., Gendron, J.M., Jonassen, E.M., Lillo, C., DeLong, A., Burlingame, A.L., Sun, Y., and Wang, Z.Y.** (2011). PP2A activates brassinosteroid-responsive gene expression and plant growth by dephosphorylating BZR1. *Nat Cell Biol* **13**, 124-131.
- Wang, Q., Jiang, M., Isupov, M.N., Chen, Y., Littlechild, J.A., Sun, L., Wu, X., Wang, Q., Yang, W., Chen, L., Li, Q., and Wu, Y.** (2020). The crystal structure of Arabidopsis BON1 provides insights into the copine protein family. *Plant J* **103**, 1215-1232.
- Wang, X., Kota, U., He, K., Blackburn, K., Li, J., Goshe, M.B., Huber, S.C., and Clouse, S.D.** (2008). Sequential transphosphorylation of the BRI1/BAK1 receptor kinase complex impacts early events in brassinosteroid signaling. *Dev Cell* **15**, 220-235.
- Zhu, J.Y., Li, Y., Cao, D.M., Yang, H., Oh, E., Bi, Y., Zhu, S., and Wang, Z.Y.** (2017). The F-box Protein KIB1 Mediates Brassinosteroid-Induced Inactivation and Degradation of GSK3-like Kinases in Arabidopsis. *Mol Cell* **66**, 648-657 e644.
- Zou, B., Hong, X., Ding, Y., Wang, X., Liu, H., and Hua, J.** (2016). Identification and analysis of copine/BONZAI proteins among evolutionarily diverse plant species. *Genome* **59**, 565-573.

Reviewer #3 (Remarks to the Author):

Copine proteins are a group of broadly presented proteins in eukaryotes, including plants and animals. BONZAI (BON) proteins are plasma membrane associated copine proteins found in plants. They were initially identified as negative regulators in plant defense responses, because *bon1* (also named *cpn1*) mutants show autoimmune phenotypes, including cell death and seedling lethality. Further analyses indicated that the autoimmune phenotypes rely on the presence of TNL-type R genes. In this manuscript, the authors studied the functions of BON homologous proteins in maize (no TNL-type R genes in maize) and found a *Zmbon1* mutant does not show autoimmune phenotypes but exhibits a dwarfed phenotype similar to the defective phenotype of *BRI1*, the receptor of brassinosteroids (BRs). The authors did a number of experiments, including a root inhibition analysis and the downstream gene expression in response to eBL treatment and confirmed that the BR signaling pathway in *Zmbon1* is indeed impaired. The authors also generated a *bon1-1 bon2-2 bon3-3 pad4-1* quadruple mutant in which all three BON genes in Arabidopsis are knocked out and the autoimmune response pathway is blocked. The quadruple mutant again showed dwarfed phenotype similar to a weak *BRI1* mutant phenotype. Overall, I find the research is interesting. But I do have a number of suggestions and concerns listed here. I hope the authors will find them useful to improve this manuscript.

1) The major concern is that the detailed molecular mechanism of BON1 in regulating BR signaling is lacking in the current version of the manuscript. The authors should analyze whether the interactions between BON1 and BAK1 or BRI1 is due to the alteration of protein stability (for example, the abundance of BRI1 and BAK1), or protein trafficking during the maturation of two RLKs in the endomembrane system, or binding affinity between BRI1 and BAK1, etc. To publish a paper in a good journal like Nature Communications, a step forward analysis is needed.

Response: Thank you for the comments and suggestions. We have carried out additional experiments to draw more clear conclusions.

First, we compared the abundance of BRI1 and BAK1 protein in *pad4-1* and *bon1-1 bon2-2 bon3-3 pad4-1* without eBL treatment and at different time points after eBL treatment, the protein levels were not altered in *bon1-1 bon2-2 bon3-3 pad4-1* compared to that of *pad4-1* (Extended Data Fig.14, Page 10-11, line 276-283), indicating that the attenuated interactions between BON1 and BAK1 or BRI1 in the quadruple mutant is not due to alteration of protein stability.

Second, we carefully examined the subcellular localization of BON1 and SERK proteins, BON proteins exhibit similar localization pattern with SERK proteins. Other than the typical plasma membrane-localization pattern, we also observed granule signal of BON1 and SERK proteins associated with the plasma membrane or in the cytosols (Extended Data Fig.15a, b, Page 11, line 284-290), suggesting that BON1 can be trafficking with the SERK proteins in the endomembrane system. However,

when we compared the localization pattern of SERK proteins in *pad4-1* and *bon1-1 bon2-2 bon3-3 pad4-1*, we didn't find significant differences (Extended Data Fig.15c, d, Page 11, line 290-293). To further confirm BON's effect on subcellular localization of SERK proteins, we purified the total plasma membrane proteins of *pad4-1*, *bon123pad4-1*, BKK1-FLAG/*pad4-1*, and BKK1-FLAG/*bon123pad4-1*, and detected SERK level by immunoblots. Again, no significant difference of SERK protein abundance was observed (Extended Data Fig.15e, f, Page 11, line 293-298). In sum, BON proteins didn't appear to affect the protein trafficking of SERK proteins.

Third, we performed Co-IP assay to compare the interaction intensity between BRI1 and SERK in *pad4-1* and *bon1-1 bon2-2 bon3-3 pad4-1*, there was less BRI1 proteins co-immunoprecipitated with SERK3 and SERK4 in the quadruple mutant (Fig.6a-d, Page 11-12, line 299-316), suggesting that the binding affinity of BRI1 and SERKs was largely affected by BON proteins.

In sum, our data support that BON proteins likely act as a chaperone or structural protein to promote the interaction between BRI1 and SERKs rather than affect the protein stability or trafficking of BRI1 and SERKs. The information has been added in Discussion (Page 14-15, line 389-394). We have also included other hypothesis in Discussion, for instance, BON proteins' potential role in PM-microdomain formation (Page 15, line 403-410). We also believe structural studies on BON-SERK-BRI1 complex will give us more direct answer in the future (Page 15, line 419-422).

2) I am wondering why the authors did not use WS accession because it lacks TNL-type R genes. The triple mutant should be easily generated by using a gene editing approach.

Response: Thanks for the comments.

According to previous study, there is a lack of TNL-type R gene SNC1 in the WS accession, however, there can be other TNL-type R genes existing in WS (Yang and Hua, 2004; Yang et al., 2006). The *bon1-2 bon2-1 bon3-1* mutant in WS background is seedling lethal and thus not available (Yang et al., 2006).

3) Most of the analyses were conducted in the quadruple mutant *bon1-1 bon2-2 bon3-3 pad4-1*. Complementation analyses are required to further solidify the conclusion. The authors should transform BON1, BON2, and BON3 back to the quadruple mutant (driven by their own promoters) and test whether the morphology and responsive gene expression levels are recovered.

Response: Thanks for the suggestions.

We didn't do the complementation analyses because we have generated the *bon1-1 pad4-1* (Col-0) and *bon1-2 bon3-1* (WS) double mutants, which resemble the pBON2::BON2+pBON3::BON3/*bon1-1 bon2-2 bon3-1 pad4-1* and pBON2::BON2/*bon1-2 bon2-1 bon3-1* transgenic plants respectively for complementation analyses. We did check the BR response of these mutants, no

significant difference was observed with these double mutants compared to the *pad4-1* mutant and the WS wild type controls (Extended Data Fig.9). These data solidify the conclusion that all three BON members are redundantly required for BR signaling in Arabidopsis.

4) I noticed the authors tested a number of BR responsive genes. To see the BR signal output, most people use the response of SAUR-AC1. The authors should add the expression levels of SAUR-AC1 in their tests.

Response: Thanks for the suggestions.

We have now checked the expression level of *SAUR-AC1* (Response Fig.4), which was indeed upregulated in both *pad4-1* and *bon123pad4-1* after eBL treatment. We did observe a mild reduction of *SAUR-AC1* expression in *bon1-1 bon2-2 bon3-1 pad4-1* compared to that of *pad4-1* after eBL treatment. However, the difference was not significant. As a control, we also didn't observe significant difference of *SAUR-AC1* in Col-0 and *bak1-4*, the loss-of-function mutant of *BAK1*, implying that the expression level of *SAUR-AC1* is not necessarily changed significantly when BR signaling is disturbed.

Response Fig.4 | Transcriptional level (FRKM) of *SAUR-AC1* in Col-0, *bak1-4*, *pad4-1* and *bon1-1 bon2-2 bon3-1 pad4-1* under mock and eBL treatment from RNA-seq.

5) Figure 2d, the grey and orange colors sometimes are hardly to tell apart due to the existence of red dots. It's better to change the colors.

Response: Thanks. We have modified the color to make the images more clear.

6) Figure 3a, the triple mutant not only regulates autoimmunity but also regulates plant development. The up part, therefore, is not complete.

Response: Thanks. We have revised the figure to show the effect of plant development on the lethal phenotype.

7) Figure 3c, eBL should be marked at the X axis.

Response: Thanks. We have revised the figure as suggested.

8) To confirm the specificity of the antibodies to against BRI1, BZR1, etc, negative controls should be used. Unfortunately, none of the negative controls were included in the manuscript. For example, an anti-BRI1 antibody should not have a cross reaction band in a *bri1* mutant background. Likewise, an anti-BZR1 antibody should not have a band in a *bzr1* mutant background.

Response: Thanks for the comments.

We have used the commercial antibodies used in previous publications (Hu et al., 2018; Chen et al., 2019; Lee et al., 2021). In this study, we have confirmed the specificity using the *bri1-5* mutant as control (Extended Data Fig.13). As described in the manual of the antibody (Response Fig.5), *bri1-5* is a mutant in the extracellular domain and the BRI1-5 protein is retained in the ER. The *bri1-5* plants contain less protein than the wild type. Also BRI1-5 migrates higher than wild type BRI1 in SDS-PAGE, because it carries ER-type high mannose N-glycans. That's why we could still see a weak mutant BRI1 band slightly higher than the band of wild type BRI1.

As for BZR1, we used the same antibody reported in the previous publication by our co-author Dr. Guang Qi, the BZR1 band is specific based on the WB signal between *bzr1* mutant and the WS wild type (Qi et al., 2022, Response Fig.5).

Response Fig.5 | Validation of antibody specificity for commercial α -BRI1 and α -BZR1

9) Figure 3c, Y axis should start from 0, not 10. It is wrong to use such a way to exaggerate the difference.

Response: Thanks. We have revised the figure as suggested.

10) The authors showed the interaction between BON1 and BRI1 only in a BIFC analysis. They should confirm it by using additional approaches.

Response: Thanks for the suggestions.

We have now re-performed SLC and Co-IP assay for the interaction between BON1 and BRI1 using the PM-localized protein PIN1 as a negative control. Our data confirmed that BRI1 interacted with BON1 but not with PIN1 in the SLC and Co-IP assay (Fig.5f, g).

Thanks again for all of your comments. We wish your concerns have been addressed.

Reference

- Chen, W., Lv, M., Wang, Y., Wang, P.-A., Cui, Y., Li, M., Wang, R., Gou, X., and Li, J.** (2019). BES1 is activated by EMS1-TPD1-SERK1/2-mediated signaling to control tapetum development in *Arabidopsis thaliana*. *Nature Communications* **10**.
- Hu, C., Zhu, Y., Cui, Y., Cheng, K., Liang, W., Wei, Z., Zhu, M., Yin, H., Zeng, L., Xiao, Y., Lv, M., Yi, J., Hou, S., He, K., Li, J., and Gou, X.** (2018). A group of receptor kinases are essential for CLAVATA signalling to maintain stem cell homeostasis. *Nature Plants* **4**, 205-211.
- Lee, H.-S., Choi, I., Jeon, Y., Ahn, H.-K., Cho, H., Kim, J., Kim, J.-H., Lee, J.-M., Lee, S., Bünting, J., Seo, D.H., Lee, T., Lee, D.-H., Lee, I., Oh, M.-H., Kim, T.-W., Belkhadir, Y., and Pai, H.-S.** (2021). Chaperone-like protein DAY plays critical roles in photomorphogenesis. *Nature Communications* **12**.
- Qi, G., Chen, H., Wang, D., Zheng, H., Tang, X., Guo, Z., Cheng, J., Chen, J., Wang, Y., Bai, M.Y., Liu, F., Wang, D., and Fu, Z.Q.** (2021). The BZR1-EDS1 module regulates plant growth-defense coordination. *Mol Plant* **14**, 2072-2087.
- Yang, H., Li, Y., and Hua, J.** (2006). The C2 domain protein BAP1 negatively regulates defense responses in *Arabidopsis*. *Plant J* **48**, 238-248.
- Yang, S., and Hua, J.** (2004). A haplotype-specific Resistance gene regulated by BONZAI1 mediates temperature-dependent growth control in *Arabidopsis*. *Plant Cell* **16**, 1060-1071.

REVIEWERS' COMMENTS

Reviewer #1 (Remarks to the Author):

The reviewers have generally resolved my questions. This is an interesting story that presents the new function of copine proteins. I have a few more suggestions for this manuscript, especially for the data presentations.

1) Since the authors used wide ranges of eBL for individual experiments (from 2 nM to 2 μ M), the authors should indicate the eBL concentrations in corresponding figure legends and methods, and explain briefly why they chose these concentrations in the main text or methods. It must be noted that " μ M" should not be simplified as "uM", either in figures or texts.

2) In Figure 4, the authors showed RNA-seq analyses without heatmaps and Venn diagrams, which makes the figure lack a general view of transcriptomic changes in the bon1bon2bon3pad4 mutants after BR treatment. If Bon proteins are required for transcriptomic changes under BR treatment, a dramatic change will be exhibited by heatmaps and Venn diagrams. In the supplemental data, it is not clear for the labels "~M1" and "~T1", and the up or down-regulated genes seem not correct. For example, in "DEGs in pad4_BL vs pad4_mock", an upregulated gene should have a higher expression after BL treatment. Besides, the whole transcriptomic data should be provided in supplemental tables, either altered or unaltered by bon mutation, or BR treatments. The recently published BR-mediated transcriptome data should be cited and discussed.

3) For the phosphoproteomic data in the supplemental table, it is not clear about the labels "Intensity T1" and "Intensity T1". Besides, the authors did not show phosphoproteomic data without BR treatment in the supplemental table, which has been discussed in the manuscript. Besides, the authors did not show an intrinsic phosphorylation control that is not affected by BR and not affected by BON mutations. Besides, the whole phosphoproteomic data should be provided in supplemental tables, either altered or unaltered by bon mutation, or BR treatments.

Response to Referee Letter

Dear reviewer,

Thank you for the comments and suggestions on the improvement of our manuscript. We have revised the manuscript comprehensively according to all of your comments. Please see the revised manuscript for details and the response to your comments as follows.

Best regards.

REVIEWERS' COMMENTS

Reviewer #1 (Remarks to the Author):

The reviewers have generally resolved my questions. This is an interesting story that presents the new function of copine proteins. I have a few more suggestions for this manuscript, especially for the data presentations.

1) Since the authors used wide ranges of eBL for individual experiments (from 2 nM to 2 μ M), the authors should indicate the eBL concentrations in corresponding figure legends and methods, and explain briefly why they chose these concentrations in the main text or methods. It must be noted that “ μ M” should not be simplified as “uM”, either in figures or texts.

Response: Thanks for the suggestions. We have revised the figures, figure legends and manuscript as suggested, and explained why different eBL concentrations were used for individual experiments in the texts (line 194-196, 227-239, 340-347, 503, and 507-508).

The concentrations of eBL were chosen according to previous reports (Liu et al., 2020; Clark et al., 2021; Montes et al., 2022). Based on our original test, we found that 2 nM, 5 nM or 10 nM eBL were all sufficient to cause obvious inhibition of root growth when Col-0, Ws, or *pad4-1* seedlings were grown on a 1/2 MS plate for 12 days. However, when Arabidopsis plants were treated with eBL by spray inoculation, higher concentration of eBL (2 μ M) was used for its better absorption by leaves in a short time (about 30 min). Our data indicate that 2 μ M eBL can induce significant changes of known BR signaling genes. Moreover, following the previous study (Kir et al., 2015), we selected 20 nM or 100 nM eBL for the root inhibition assay in maize, and found that 20 nM or 100 nM eBL was sufficient to cause obvious inhibition of root growth of the wild-type maize KN5585 or B73 (Figure 2c, d and Supplementary Fig.8).

Reference:

Liu, X. et al. Comparative transcriptomic analysis to identify brassinosteroid response

genes. *Plant Physiol.* **184**, 1072-1082 (2020).

Clark, N. et al. Integrated omics networks reveal the temporal signaling events of brassinosteroid response in Arabidopsis. *Nat. Commun.* **12**, 5858 (2021).

Montes, C. et al. Integration of multi-omics data reveals interplay between brassinosteroid and target of rapamycin complex signaling in Arabidopsis. *New Phytol.* **236**, 893-910 (2022).

Kir, G. et al. RNA interference knockdown of BRASSINOSTEROID INSENSITIVE1 in maize reveals novel functions for brassinosteroid signaling in controlling plant architecture. *Plant Physiol.* **169**, 826-839 (2015).

2) In Figure 4, the authors showed RNA-seq analyses without heatmaps and Venn diagrams, which makes the figure lack a general view of transcriptomic changes in the *bon1bon2bon3pad4* mutants after BR treatment. If Bon proteins are required for transcriptomic changes under BR treatment, a dramatic change will be exhibited by heatmaps and Venn diagrams. In the supplemental data, it is not clear for the labels “~M1” and “~T1”, and the up or down-regulated genes seem not correct. For example, in "DEGs in *pad4_BL* vs *pad4_mock*", an upregulated gene should have a higher expression after BL treatment. Besides, the whole transcriptomic data should be provided in supplemental tables, either altered or unaltered by *bon* mutation, or BR treatments. The recently published BR-mediated transcriptome data should be cited and discussed.

Response: Thanks for the suggestions.

We have included the Venn diagrams (Figure.4b) and heatmaps (Figure.4c) in Figure 4. The new data indicated that there was great overlapping of differentially expressed genes (DEGs) between Col-0 and *bak1-4* and DEGs between *pad4-1* and *bon1-1 bon2-2 bon3-3 pad4-1* upon eBL treatment, and these DEGs show similar pattern in *bak1-4* and *bon1-1 bon2-2 bon3-3 pad4-1*.

We have revised the labels in the Supplementary Table.2. We have also provided the whole transcriptomic data for Col-0, *bak1-4*, *pad4-1* and *bon1bon2bon3pad4-1* under mock or eBL treatment in Supplementary Data File.2. In addition, we have cited the recently published BR-mediated transcriptome data in the discussion (line 412-417).

3) For the phosphoproteomic data in the supplemental table, it is not clear about the labels “Intensity T1” and “Intensity T1”. Besides, the authors did not show phosphoproteomic data without BR treatment in the supplemental table, which has been discussed in the manuscript. Besides, the authors did not show an intrinsic phosphorylation control that is not affected by BR and not affected by BON mutations. Besides, the whole phosphoproteomic data should be provided in supplemental tables, either altered or unaltered by *bon* mutation, or BR treatments.

Response: Thanks for the suggestions.

We have revised the labels and provided the differentially phosphorylated proteins between *pad4-1* and *bon1bon2bon3pad4-1* without BR treatment (mock) in the Supplementary Table.3. Besides, the whole phosphoproteomics data of *pad4-1* and *bon1bon2bon3pad4-1* under mock or eBL treatment have been provided in the Supplementary Data File.3.

Regrettably, we did not send the samples of Col-0 and *bak1-4* for phosphoproteomics due to the high cost. However, we used *pad4-1* as the control of *bon1bon2bon3pad4-1* since BR signaling didn't appear to be affected by the *pad4-1* mutation (Supplementary Fig. 9b, c and Supplementary Fig.10c). The phosphoproteomics data indicated that phosphorylation levels of multiple BR signaling components including BRI1, SERKs, BKI1, BSKs and BSLs were increased after eBL treatment in *pad4-1*, implying that the phosphorylation cascade in BR signaling was activated in *pad4-1* (Supplementary Fig. 16c and Supplementary Data File.3). Moreover, phosphopeptides from all above proteins were lower in *bon1bon2bon3pad4-1* than *pad4-1* upon eBL treatment (Supplementary Fig. 16c and Supplementary Data File.3). Besides, the Co-IP and immunoblot data using anti-phosphoserine or anti-phosphothreonine antibodies further confirmed the phosphoproteomics data (Fig. 6f-i, Supplementary Fig.17 and Supplementary Fig. 18)